# Interferon alpha-inducible protein 6 regulates NRASQ61K-induced melanomagenesis and growth

Romi Gupta[1], Matteo Forloni[1], Malik Bisserier[1], Shaillay Kumar Dogra[2], Qiaohong Yang[1], Narendra Wajapeyee[1]*

[1]Department of Pathology, Yale University School of Medicine, New Haven, United States; [2]Singapore Institute of Clinical Sciences, Agency for Science Technology and Research (A*STAR), Brenner Center for Molecular Medicine, Singapore, Singapore

**Abstract** Mutations in the *NRAS* oncogene are present in up to 20% of melanoma. Here, we show that interferon alpha-inducible protein 6 (IFI6) is necessary for NRASQ61K-induced transformation and melanoma growth. *IFI6* was transcriptionally upregulated by NRASQ61K, and knockdown of *IFI6* resulted in DNA replication stress due to dysregulated DNA replication via E2F2. This stress consequentially inhibited cellular transformation and melanoma growth via senescence or apoptosis induction depending on the RB and p53 pathway status of the cells. NRAS-mutant melanoma were significantly more resistant to the cytotoxic effects of DNA replication stress-inducing drugs, and knockdown of *IFI6* increased sensitivity to these drugs. Pharmacological inhibition of *IFI6* expression by the MEK inhibitor trametinib, when combined with DNA replication stress-inducing drugs, blocked NRAS-mutant melanoma growth. Collectively, we demonstrate that IFI6, via E2F2 regulates DNA replication and melanoma development and growth, and this pathway can be pharmacologically targeted to inhibit NRAS-mutant melanoma.

**\*For correspondence:** narendra.
wajapeyee@yale.edu

**Competing interests:** The authors declare that no competing interests exist.

## Introduction

Melanoma is the deadliest form of skin cancer, accounting for ~80% of skin cancer-related deaths (*Miller and Mihm, 2006*). Some of the most commonly observed oncogenic events in melanoma are activating mutations of the oncogenic neuroblastoma RAS viral oncogene homolog (NRAS; typically NRASQ61K/R), which occur in 20% of cases (*Miller and Mihm, 2006*; *Tsao et al., 2012*). Oncogenic NRAS causes constitutive activation of NRAS, resulting in activation of multiple downstream signaling pathways. These signaling pathways promote proliferation and reduce apoptosis to facilitate cellular transformation, tumor growth, and metastatic progression (*Downward, 2003*; *Karnoub and Weinberg, 2008*; *Wellbrock et al., 2004*). Key pathways regulated by NRAS include the phosphoinositide 3-kinase (PI3K), mitogen-activated protein kinase (MAPK), and Ral guanine nucleotide dissociation stimulator (RalGDS) pathways, all of which are shown to play important roles in NRAS-driven oncogenesis (*Downward, 2003*; *Karnoub and Weinberg, 2008*).

Both the PI3K and MAPK pathways have been targeted using very effective small molecule inhibitors as a clinical approach to treat RAS-mutant cancers, including NRAS-mutant melanoma. However, other pro-survival pathways can be activated by feedback and crosstalk from the pathway being targeted. For example, inhibition of the PI3K pathway alone can result in compensatory upregulation of the MAPK pathway (*Carracedo et al., 2008*; *Serra et al., 2011*). For this and several other known and unknown reasons, these approaches have not proven clinically beneficial (*Baines et al., 2011*). Thus, new therapeutic approaches for treating NRAS-mutant melanoma and other RAS-mutant cancers are needed.

To discover and develop more effective approaches to treat NRAS-mutant melanoma, we aimed to understand the mechanism of oncogenic NRAS-induced transformation and tumor maintenance. We identified interferon alpha-inducible protein 6 (IFI6) as a new regulator of oncogenic NRAS-induced melanocyte transformation and melanoma tumor growth, which functions by regulating E2F2 expression and consequently DNA replication. Surprisingly, we also found that NRAS-mutant melanoma cells were significantly more resistant to the drugs that induce DNA replication stress than were BRAF-mutant, NF1-deficient, or triple wild-type melanomas. In addition, treatment with the MEK inhibitor trametinib, which reduces IFI6 expression, when combined with drugs that induce DNA replication stress, potently inhibited NRAS-mutant melanoma tumor growth in cell culture and in mice. Taken together, our results identify a new role for IFI6 in E2F2-mediated regulation of DNA replication and melanoma development and growth. These studies also uncover a pharmacologically tractable DNA replication stress resistance pathway that can be targeted to inhibit NRAS-mutant melanoma.

## Results

### Oncogenic NRAS transcriptionally upregulates *IFI6* via MAPK pathway

Oncogenic mutations in neuroblastoma RAS (NRAS), typically in codon 61, are observed in <~20% of melanoma (2015). However, NRAS-mutant melanoma currently lacks effective targeted therapies, and targeting the pro-survival pathways downstream of oncogenic NRAS (e.g., PI3K or MEK inhibitors) have not been successful (*Britten, 2013*; *Samatar and Poulikakos, 2014*; *Zhao and Adjei, 2014*). Thus, a better understanding of NRAS-mutant melanoma is required for developing effective targeted therapies. Toward this end, we sought to identify factors that are necessary for oncogenic NRAS-induced melanocyte transformation and melanoma growth.

First, we performed transcriptome-wide gene expression analyses. To do so, we transformed immortalized melanocytes (MEL-ST cells) using oncogenic NRAS, NRASQ61K (hereafter referred to as MEL-ST/NRASQ61K), and then we analyzed the gene expression changes using an Illumina gene expression array. Our gene expression data analyses identified 301 genes that were significantly upregulated (p<0.05, fold-change >2.0) in MEL-ST/NRASQ61K cells compared to MEL-ST cells with an empty vector control (*Supplementary file 1A* and *Figure 1—figure supplement 1*). Among the top five genes were *IL8* and *IL1B*, which were previously implicated in RAS-mediated transformation (*Cataisson et al., 2012*; *Sparmann and Bar-Sagi, 2004*). However, we also identified three interferon-stimulated genes (ISGs)—*IFI6, IFI27*, and *MX1*—that have not been previously implicated in oncogenic NRAS-induced transformation and melanoma tumor growth (*Figure 1A* and *Figure 1—figure supplement 2*). Furthermore, an analysis of previously published melanoma datasets revealed that *IFI6* is overexpressed in melanoma samples (*Figure 1B–C*) (*Barretina et al., 2012*; *Haqq et al., 2005*; *Riker et al., 2008*; *Talantov et al., 2005*). Based on these results, we focused our studies on IFI6.

First, we determined the mechanism by which NRASQ61K transcriptionally upregulates the expression of *IFI6*. Toward this end, we employed RAS mutants that specifically activate either the MAPK (HRAS v12 S35) or PI3K pathway (HRAS v12 C40). Using these mutants, we found that the MAPK pathway stimulated *IFI6* expression effectively in MEL-ST cells (*Figure 1D–E*). To confirm this finding, we used the constitutively active MEK construct MEK-DD (*Boehm et al., 2007*) and found that the introduction of MEK-DD in MEL-ST cells was sufficient to stimulate *IFI6* expression (*Figure 1F–G*). Finally, we analyzed the expression of *IFI6* and key MAPK transcriptional targets in 20 patient-derived melanoma samples. We observed that *IFI6* expression strongly correlated with the expression of other known MAPK transcriptional targets (*Figure 1H*). Additionally, IFI6 overexpression significantly correlated with the NRAS mutation status in patient-derived melanoma samples (*Figure 1I*) (*Haqq et al., 2005*). These results demonstrate that NRASQ61K activates *IFI6* expression through the MAPK pathway.

In melanoma, the MAPK pathway can also be activated as a result of mutations in BRAF genes (e.g., BRAFV600E) or loss of neurofibromatosis type 1 (NF1) activity due to inactivating mutations (*Coverley et al., 2002*; *Davies et al., 2002*; *Krauthammer et al., 2015*). Therefore, we asked whether BRAFV600E or *NF1* knockdown could result in the transcriptional upregulation of *IFI6*, similar to NRASQ61K. To this end, we either introduced BRAFV600E or knocked down *NF1* expression

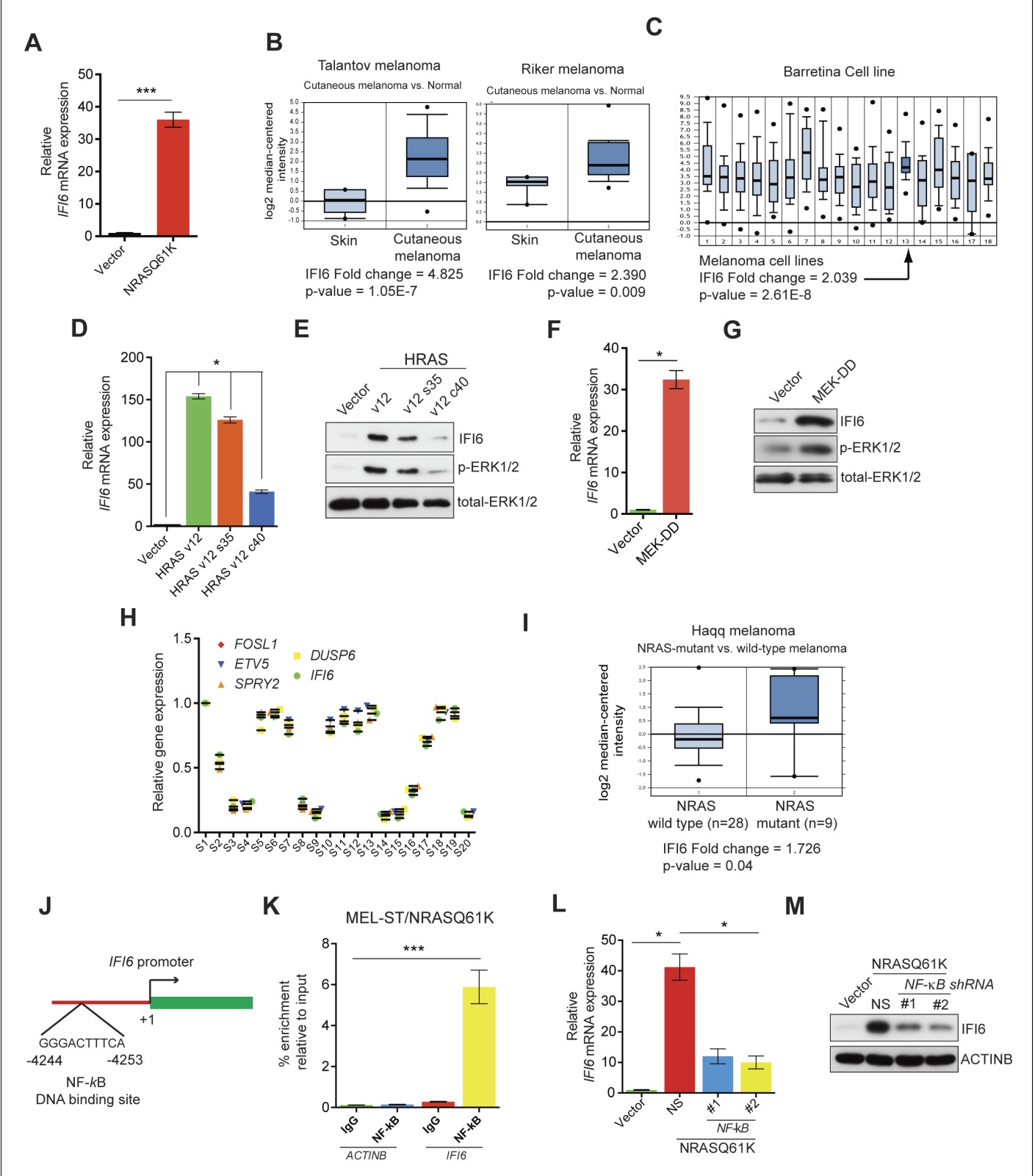

**Figure 1.** *IFI6* is transcriptionally upregulated by NRASQ61K via MAPK pathway. (**A**) Relative *IFI6* mRNA expression in MEL-ST/NRASQ61K cells compared to empty vector-expressing MEL-ST cells. (**B**) Box plots for *IFI6* mRNA expression in indicated melanoma gene expression datasets show significantly higher *IFI6* mRNA expression in patient-derived cutaneous melanoma samples (2) compared to normal skin controls (1). (**C**) Box plot for *IFI6* mRNA expression in Barretina cell line dataset. Lane 13 shows average of IFI6 mRNA expression in melanoma cell lines. (**D**) MEL-ST cells expressing

*Figure 1 continued on next page*

*Figure 1 continued*

empty vector or indicated HRAS mutants were analyzed by RT-qPCR. The relative expression of *IFI6* mRNA in HRAS mutant-expressing MEL-ST cells compared to empty vector-expressing MEL-ST cells. (E) Relative expression of indicated proteins was evaluated by immunoblotting in MEL-ST cells expressing an empty vector or indicated HRAS mutants. (F) MEL-ST cells expressing empty vector or MEK-DD were analyzed for *IFI6* mRNA expression by RT-qPCR. *IFI6* mRNA expression in MEK-DD expressing MEL-ST cells relative to empty vector is shown. (G) MEL-ST cells expressing empty vector or MEK-DD were analyzed for indicated proteins by immunoblotting. (H) Analysis of patient-derived melanoma samples (n = 20) reveals co-expression of *IFI6* with MAPK target genes. (I) Box plot for indicated melanoma gene expression dataset shows significantly higher *IF6* mRNA expression in patient-derived NRAS-mutant melanoma samples (2) compared to NRAS wild-type melanoma samples (1). (J) Schematic presentation of NF-κB DNA binding site on the *IFI6* promoter. (K) MEL-ST/NRASQ61K cells were analyzed for NF-κB enrichment using the ChIP assay.% NF-κB enrichment in comparison to IgG on the *ACTIN* or *IFI6* gene promoter is shown. (L) MEL-ST cells expressing empty vector or NRASQ61K with NS or *NF-κB* shRNAs were analyzed for *IFI6* mRNA expression by RT-qPCR. Relative IFI6 mRNA in comparison to empty vector expressing MEL-ST cells is shown. (M) MEL-ST cells expressing empty vector or NRASQ61K with NS or *NF-κB* shRNAs were analyzed for IFI6 protein levels by immunoblotting. ACTINB served as the loading control. In all panels, data are presented as mean ± SEM, and *$p<0.05$ and ***$p<0.0005$.

The following figure supplements are available for figure 1:

**Figure supplement 1.** NRASQ61K transcriptionally upregulates interferon-stimulated genes.

**Figure supplement 2.** Regulation of MX1 and IFI6 by NRASQ61K.

**Figure supplement 3.** Monitoring regulation of IFI6 by BRAFV600E and NF1 loss.

**Figure supplement 4.** STAT1 is not necessary for IFI6 expression in MEL-ST/NRASQ61K cells.

**Figure supplement 5.** Oncogenic NRASQ61K transcriptionally upregulates *IFI6* via transcription factor NF-κB.

**Figure supplement 6.** IKKβ is necessary for NF-κB activation and NRASQ61K-induced IFI6 upregulation.

in MEL-ST cells (*Figure 1—figure supplement 3*). As controls, we used empty vector or non-specific (NS) small hairpin RNA (shRNA), respectively. These cells were then analyzed for *IFI6* expression by RT-qPCR and immunoblot analysis. Our results showed that BRAFV600E, similar to NRASQ61K, was able to activate IFI6 expression. However, *NF1* knockdown did not result in *IFI6* upregulation (*Figure 1—figure supplement 3*). These results indicate that *NF1* loss is not functionally equivalent to BRAFV600E or NRASQ61K regarding its ability to activate *IFI6* expression.

Next, we asked which transcription factors downstream of the MAPK pathway were necessary to activate expression of *IFI6*. Toward this end, we analyzed the promoter of *IFI6* using rVISTA2.0 (*Loots and Ovcharenko, 2004*) and identified DNA binding sites for transcription factors NF-κB and STAT1 (*Figure 1J* and *Figure 1—figure supplement 4*). To test if NF-κB or STAT1 directly regulate *IFI6* transcription, we first performed a chromatin immunoprecipitation (ChIP) assay. MEL-ST/ NRASQ61K cells showed enrichment of NF-κB on the *IFI6* promoter relative to MEL-ST cells expressing an empty vector (*Figure 1K*). However, we did not observe enrichment for STAT1 on the *IFI6* promoter in MEL-ST/NRASQ61K cells relative to MEL-ST cells expressing an empty vector (*Figure 1—figure supplement 4*). To further test whether NF-κB and STAT1 influence *IFI6* mRNA expression, we measured the expression of *IFI6* in MEL-ST/NRASQ61K cells after knocking down the expression of either *NF-κB (p65/ RelA)* or *STAT1*. We found that *NF-κB* knockdown markedly decreased *IFI6* expression in MEL-ST/NRASQ61K cells (*Figures 1L–M* and *Figure 1—figure supplement 5*), whereas *STAT1* knockdown had no effect (*Figure 1—figure supplement 5*). We also determined the mechanism of NF-κB activation downstream of NRASQ61K. To this end, we knocked down the expression of *IKKβ*, a kinase that phosphorylates and inactivates IκB; IκB inhibits NF-κB by preventing its nuclear localization (*Li et al., 1999a*, *1999b*). IKKβ was previously shown to be necessary for NF-κB activity and HRASv12-driven melanoma growth in mice (*Yang et al., 2010*). Knockdown of *IKKβ* resulted in reduced phosphorylation of IκB (*Figure 1—figure supplement 6*) and reduced NF-κB reporter activity (*Figure 1—figure supplement 6*). Analysis of IFI6 also revealed that *IKKβ* knockdown attenuated NRAS-induced *IFI6* expression (*Figure 1—figure supplement 6*). ChIP analysis showed decreased enrichment of NF-κB on the *IFI6* promoter upon *IKKβ* knockdown

(*Figure 1—figure supplement 6*). Collectively, these results demonstrate that NRASQ61K, in a MAPK pathway-dependent manner via NF-κB, stimulates the transcription of *IFI6*.

## IFI6 is required for NRASQ61K-induced transformation and NRAS-mutant melanoma tumor growth

Next, we asked if IFI6 is necessary for NRASQ61K-induced transformation. Toward this end, we knocked down the expression of *IFI6* using shRNA in MEL-ST cells, and then we introduced the NRASQ61K mutant via lentiviral infection. These cells were tested for their ability to form colonies in soft agar and tumors in mice. We found that *IFI6* knockdown inhibited the ability of NRASQ61K to transform MEL-ST cells, as shown by the reduced colony formation in soft agar (*Figure 2A–B* and *Figure 2—figure supplement 1*) and tumor formation in mice (*Figure 2C*). We also performed rescue experiments by expressing shRNA that targets the 3'-UTR of *IFI6*. Our results show that ectopic expression of shRNA-resistant *IFI6* open reading frame (ORF) rescued growth both in soft agar and in mice (*Figure 2—figure supplement 2*). Notably, *NF-κB* knockdown also resulted in the inhibition

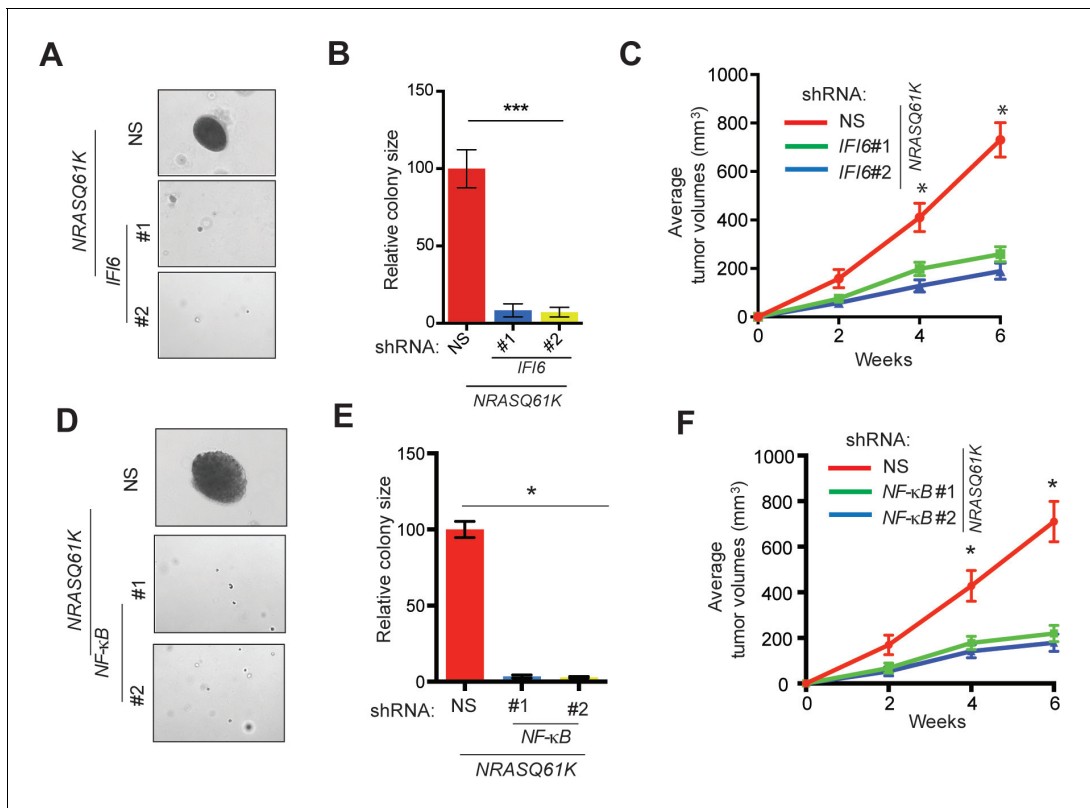

**Figure 2.** IFI6 is necessary for NRASQ61K-induced melanocyte transformation. (**A**) MEL-ST/NRASQ61K cells expressing *IFI6* or non-specific (NS) shRNA were analyzed for colony-forming potential using a soft agar assay. Representative images are shown. (**B**) Relative colony size for the soft-agar assay presented in panel **A** is shown. (**C**) MEL-ST/NRASQ61K cells expressing *IFI6* or non-specific (NS) shRNA were injected subcutaneously into the flanks of athymic nude mice. Average tumor volumes (n = 5) for MEL-ST/NRASQ61K cells expressing NS or *IFI6* shRNA are shown at the indicated time points. (**D**). MEL-ST/NRASQ61K cells expressing NS or *NF-κB* shRNA were analyzed for colony-forming potential using a soft agar assay. Representative images are shown. (**E**) Relative colony size for the soft-agar assay presented in panel **D** is shown. (**F**) MEL-ST/NRASQ61K cells expressing NS or *NF-κB* shRNA were injected subcutaneously into the flanks of athymic nude mice. Average tumor volumes (n = 5) for MEL-ST/NRASQ61K cells expressing NS or *NF-κB* shRNA are shown at the indicated time points. In all panels, data are presented as mean ± SEM, and *p<0.05 and ***p<0.0005.

The following figure supplements are available for figure 2:

**Figure supplement 1.** Monitoring expression of IFI6 and E2F2 proteins following *IFI6* knockdown.

**Figure supplement 2.** Ectopic expression of shRNA-resistant *IFI6* ORF rescues tumor growth in MEL-ST/NRASQ61K cells expressing *IFI6* shRNA.

of NRASQ61K-induced transformation, further supporting a role for NF-κB in *IFI6* expression (*Figure 2D–F* and *Figure 1—figure supplement 5*).

Based on these results, we asked if IFI6 is necessary for NRAS-mutant melanoma-driven tumor growth. Toward this end, we knocked down the expression of *IFI6* in three NRAS-mutant melanoma cell lines (YUGASP, M318, and SKMEL-103) and one NRAS wild-type melanoma cell line (YUVON) and monitored their colony- and tumor-forming ability. Our results show that shRNA-induced knockdown of *IFI6* inhibited the ability of NRASQ61K mutant melanoma cells to form colonies in soft agar (*Figures 3A–B*, *Figure 2—figure supplement 1* and *Figure 3—figure supplement 1*) and to form tumors in mice (*Figure 3C*). However, *IFI6* knockdown did not affect colony or tumor formation in YUVON cells (*Figure 3D–F*). We also performed rescue experiments in YUGASP cells by expressing shRNA that targets the 3' untranslated region (UTR) of *IFI6*. Our results show that ectopic expression of shRNA-resistant *IFI6* open reading frame (ORF) was able to rescue growth in both soft agar and in mice (*Figure 3—figure supplement 2*). Collectively, these results demonstrate that *IFI6* is necessary for NRASQ61K-induced transformation and NRAS-mutant melanoma growth.

## IFI6 loss results in dysregulated DNA replication via transcription factor E2F2 and inhibition of NRAS-mutant melanoma tumor growth

After confirming the role of IFI6 in the regulation of melanocyte transformation and melanoma growth, we asked how IFI6 regulates these phenotypes. To answer this question, we knocked down the expression of *IFI6* in the NRAS-mutant melanoma cell line YUGASP and performed a microarray analysis (*Supplementary file 1B and 1C*). We found that the loss of *IFI6* expression resulted in increased expression of transcription factor *E2F2* and its target genes (*Figure 4A* and *Figure 2—figure supplement 1*), many of which were previously documented to regulate DNA replication (e.g., *CCNE1*, *MCM3*, *MCM10*) (*Aladjem, 2007*; *Coverley et al., 2002*; *Thu and Bielinsky, 2013*). We observed the same results in two additional NRAS-mutant melanoma cell lines, M318 and SKMEL-103 (*Figure 4—figure supplement 1* and *Figure 2—figure supplement 1*). Furthermore, consistent with the increased level of *E2F2* mRNA, we observed increased E2F2 protein level and increased E2F2 enrichment on the promoter of its target gene *MCM10* (*Figure 2—figure supplement 1* and *Figure 4—figure supplement 2*). Analysis of other E2F family genes in NRAS-mutant cells, MEL-ST/NRASQ61K cells and primary human melanocytes after *IFI6* knockdown revealed no changes in the expression of other E2F genes (*Figure 4—figure supplement 3*). Based on these findings, we hypothesized that the loss of *IFI6* results in dysregulated DNA replication via E2F2 upregulation, which in turn blocks NRAS-mutant melanoma tumor growth and melanomagenesis. To test this likelihood, we analyzed the DNA content of YUGASP cells expressing either *IFI6* or NS shRNA by fluorescence-activated cell sorting (FACS). YUGASP cells expressing *IFI6* shRNA displayed a significantly higher percentage of cells in S phase than the cells expressing NS shRNA (*Figure 4B* and *Figure 4—figure supplement 4*). Next, to test if *IFI6* knockdown results in dysregulation of the DNA replication process, we used a DNA fiber assay to directly measure DNA replication (*Merrick et al., 2004*). Toward this end, we analyzed YUGASP cells expressing *IFI6* or NS shRNA. Our results showed that the loss of *IFI6* resulted in dysregulated DNA replication, as observed by significantly fewer ongoing DNA replication forks, significantly more stalled forks, and a significantly higher number of newly fired origins of replication compared to melanoma cells expressing NS shRNA (*Figures 4C–D* and *Figure 4—figure supplement 4*). This result also explains the high number of cells in S phase of the cell cycle.

Previous studies have shown that dysregulated DNA replication can result in senescence induction (*Bartkova et al., 2006*; *Di Micco et al., 2006*). Therefore, we asked if *IFI6* knockdown results in dysregulated DNA replication-induced cellular senescence, which may consequentially inhibit melanoma growth. To test this, we knocked down *IFI6* in NRAS-mutant melanoma cell lines, YUGASP, M318, and SKMEL-103, and measured well-accepted markers of cellular senescence including senescence-associated β-gal (SA-β-gal) and acetylated histone H3K9 (H3K9Ac) (*Narita et al., 2006*, *2003*). *IFI6* knockdown resulted in the accumulation of markers of cellular senescence (*Figure 4E–G*), suggesting that senescence was induced. In addition, we also measured the phosphorylated histone γH2A.X as a marker of DNA damage, which showed that DNA replication stress-induced DNA damage was upregulated (*Figure 4G*).

Next, we asked if the *E2F2* upregulation that results from *IFI6* knockdown is necessary for melanoma growth inhibition. To determine this, we simultaneously knocked down the expression of *E2F2*

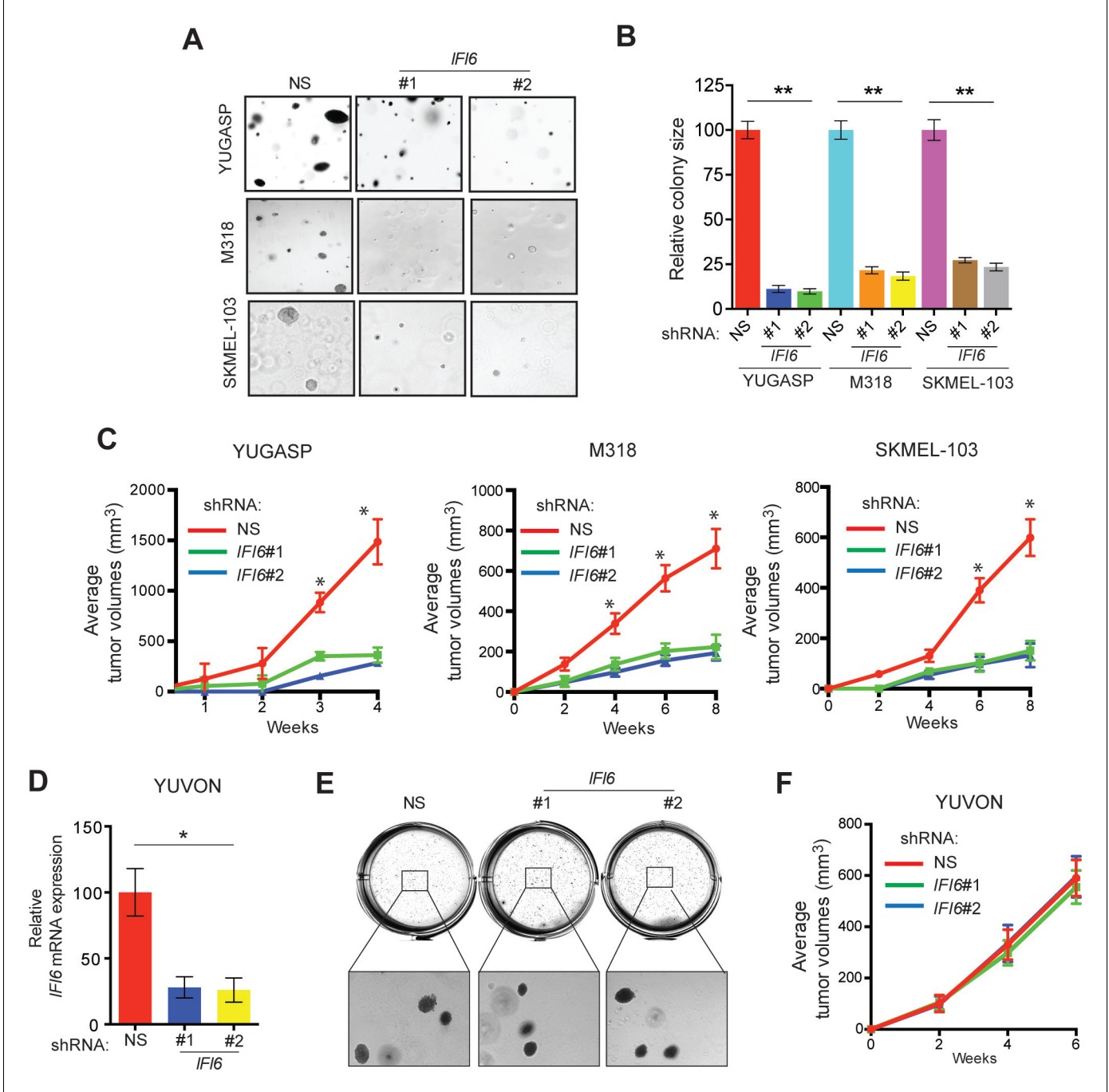

**Figure 3.** IFI6 is necessary for NRAS-mutant melanoma tumor growth. (**A**) NRAS-mutant melanoma cell lines expressing either non-specific (NS) or *IFI6* shRNA were analyzed for anchorage-independent growth using the soft agar assay. Representative images for the indicated melanoma cell lines expressing NS or *IFI6* shRNA are shown. (**B**) Relative colony size in the soft agar assay presented in panel **A** are plotted for the indicated melanoma cell lines expressing either NS or *IFI6* shRNA. (**C**) The indicated NRAS-mutant melanoma cell lines expressing either NS or *IFI6* shRNA were injected subcutaneously into the flank of athymic nude mice. Average tumor volumes (n = 5) at the indicated time points are shown. (**D**) YUVON cells expressing NS or *IFI6* shRNA were analyzed for *IFI6* mRNA expression by RT-qPCR. Relative *IFI6* mRNA expression (%) in YUVON cells expressing *IFI6* shRNA in comparison to NS shRNA-expressing cells is shown. (**E**) YUVON cells expressing NS or *IFI6* shRNA were analyzed for their ability to grow in soft agar. Representative wells of YUVON cells expressing the indicated shRNA and microscopic images are shown. (**F**) YUVON cells expressing NS or *IFI6* shRNA were injected subcutaneously in athymic nude mice. Average tumor volumes (n = 5) for the indicated time points are shown. In all panels, data are presented as mean ± SEM, and *p<0.05 and **p<0.005.

The following figure supplements are available for figure 3:

*Figure 3 continued on next page*

*Figure 3 continued*

**Figure supplement 1.** IFI6 is necessary for NRAS-mutant melanoma cell growth.

**Figure supplement 2.** Ectopic expression of shRNA-resistant *IFI6* ORF rescues tumor growth in YUGASP cells expressing *IFI6* shRNA.

and *IFI6* using shRNA (*Figure 5A–B*) and observed a significant rescue of the NRAS-mutant melanoma cell's ability to form colonies in soft agar (*Figure 5C–D*) and tumors in mice (*Figure 5E–F*). Furthermore, simultaneous knockdown of *E2F2* and *IFI6* in melanoma cells resulted in a lower percentage of cells in S phase and normalized DNA replication compared with cells expressing *IFI6* shRNA (*Figure 6A–B* and *Figure 4—figure supplement 4*). We also evaluated senescence markers in this scenario and found that simultaneous knockdown of *E2F2* and *IFI6* prevented IFI6-induced senescence in melanoma cells (*Figure 6C–E*). The shRNA against *E2F2* was highly specific to *E2F2*, as demonstrated by the unchanged mRNA levels of other E2F family genes (*Figure 6—figure supplement 1*).

Further studies revealed that the introduction of NRASQ61K in primary human melanocytes increased *IFI6* expression and downregulated *E2F2* and several E2F2 target genes, similar to melanoma cells (*Figure 7A*). Additionally, knockdown of *IFI6* in primary human melanocytes resulted in the dysregulated DNA replication (*Figure 7B* and *Figure 7—figure supplement 1*). Furthermore, *IFI6* knockdown in primary human melanocytes enhanced NRASQ61K-induced senescence, as demonstrated by markers of cellular senescence (SA-β-gal and H2K9Ac) (*Figure 7C–E*). Consistent with our earlier results we observed an increase in γH2A.X upon *IFI6* knockdown (*Figure 7E*).

Based on these results, we tested whether immortalized melanocytes expressing NRASQ61K behaved similarly to the NRAS-mutant melanoma cells and primary human melanocytes that show regulation of E2F2 and its target genes. In complete agreement with our previous results, expression of NRASQ61K resulted in increased *IFI6* expression and decreased expression of E2F2 and its target genes in MEL-ST cells (*Figure 8A*). We also observed dysregulated DNA replication in NRASQ61K-transformed MEL-ST cells expressing *IFI6* shRNA (*Figure 8B* and *Figure 8—figure supplement 1*). However, senescence induction was not observed in MEL-ST/NRASQ61K cells expressing *IFI6* shRNA (*Figure 8—figure supplement 2*). This is consistent with the fact that the MEL-ST cells express the SV40 early region (SV40-ER) and thus inactivate both p53 and retinoblastoma tumor suppressor (RB) pathways. However, previous studies have shown that uncontrolled DNA replication-induced DNA damage can result in apoptosis induction even in the absence of p53 and RB (*Aladjem et al., 1998*; *Knudsen et al., 2000*; *Strasser et al., 1994*). Therefore, we asked whether loss of *IFI6* results in apoptosis induction when p53 and RB are inhibited. To this end, we tested anchorage-independent cell growth in plates coated with poly(2-hydroxyethyl methacrylate) (poly-HEMA) and using the recently developed growth in low attachment (GILA) assay (*Rotem et al., 2015*). We found that loss of *IFI6* in MEL-ST/NRASQ61K cells increased apoptosis under anchorage-independent growth conditions compared to NS shRNA-expressing cells (*Figure 8C–D*). We confirmed these results by annexin V staining and immunoblot analysis of cleaved caspase 3 (*Figure 8E–F*). Taken together, our results demonstrate that upon p53 and RB pathway inactivation, *IFI6* loss results in the upregulation of *E2F2* and its target genes involved in DNA replication. This, in turn, causes DNA replication stress and consequent DNA damage, thereby inducing apoptosis. However, when the p53 and/or RB pathway is intact, *IFI6* loss results in the induction of cellular senescence. This consequentially inhibits mutant NRAS-induced melanocyte transformation and melanoma growth.

## NRAS-mutant melanoma are resistant to DNA replication stress-inducing agents

Finally, we asked whether these results are of clinical significance. Toward this end, we tested the response of NRAS-mutant melanoma cells to drugs that induce cytotoxicity by inducing DNA replication stress. We treated NRAS-mutant (YUGASP, SKMEL-2, SKMEL-103, and M318), BRAF-mutant (SKMEL-28 and A375), NF1-mutant/null (MeWo and YUTOGS), and NRAS/BRAF/NF1 wild-type (YUVON) melanoma cells with three DNA replication stress-inducing agents: aphidicolin,

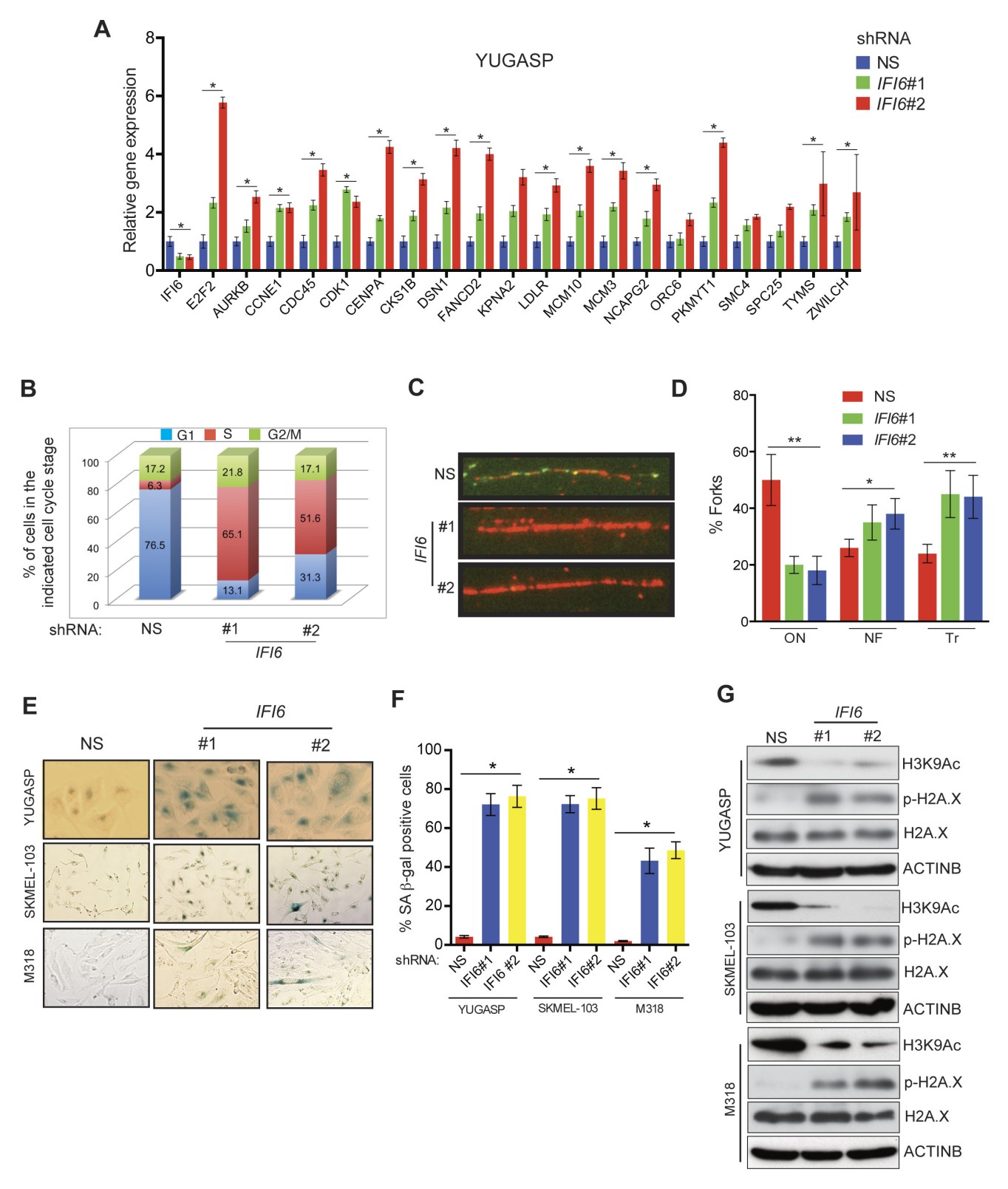

**Figure 4.** *IFI6* loss results in E2F2-mediated dysregulation of DNA replication and induction of cellular senescence in melanoma cells. (**A**) YUGASP cells expressing either *IFI6* or non-specific (NS) shRNA were analyzed by RT-qPCR. Expression of each gene was calculated in *IFI6* knockdown cells relative to NS shRNA-expressing cells. (**B**) FACS analysis of YUGASP cells expressing NS or *IFI6* shRNA. The percentage of cells in S phase is shown in red. (**C**) YUGASP cells expressing NS or *IFI6* shRNA were analyzed using a DNA fiber assay. Representative images of the DNA fibers are shown. (**D**) YUGASP

*Figure 4 continued on next page*

*Figure 4 continued*

cells expressing NS or *IFI6* shRNA were analyzed using the DNA fiber assay. Percentages of ongoing (ON), newly fired (NF), and terminated (Tr) DNA forks are shown. (**E**) YUGASP, SKMEL-103, and M318 cells expressing NS or *IFI6* shRNA were analyzed for SA-β-gal activity. Representative images of cells stained for SA-β-gal activity for the indicated melanoma cell lines expressing NS or *IFI6* shRNA are shown. (**F**) Percentage of SA-β-gal-positive cells for the experiment shown in panel **E** for the indicated melanoma cell lines expressing NS or *IFI6* shRNA is plotted. (**G**) YUGASP, SKMEL-103 and M318 cells expressing each shRNA were analyzed for H3K9Ac, γH2A.X, and H2A.X using immunoblot. ACTINB was used as a loading control. In all panels, data are presented as mean ± SEM, and *p<0.05, **p<0.005.

The following figure supplements are available for figure 4:

**Figure supplement 1.** *IFI6* knockdown results in upregulation of *E2F2* and its target genes.

**Figure supplement 2.** Increased enrichment of E2F2 on *MCM10* promoter following *IFI6* knockdown.

**Figure supplement 3.** *IFI6* knockdown affects expression of E2F2 but not the expression of other E2F family genes.

**Figure supplement 4.** Simultaneous knockdown of IFI6 and E2F2 restores DNA replication defect.

camptothecin, and hydroxyurea (*Durkin et al., 2008*; *O'Connell et al., 2010*; *Petermann et al., 2010*). We found that NRAS-mutant melanoma cells were significantly more resistant to the cytotoxic effects of DNA replication stress-inducing drugs compared to other genotypes (BRAF-mutant, NF1-deficient, or triple wild-type) (*Figure 9A*). Furthermore, we noted that the knockdown of *IFI6* sensitized NRAS-mutant cells to DNA replication stress-inducing agents (*Figure 9B*), and this effect was rescued by simultaneous knockdown of both *E2F2* and *IFI6* (*Figure 9B*). Because we had observed that the MAPK pathway regulates IFI6, we tested whether pharmacological MEK inhibitors could be combined with DNA replication stress-inducing drugs to treat NRAS-mutant melanoma. Remarkably, we found that simultaneous treatment of NRAS-mutant cells with the MEK inhibitor trametinib and either hydroxyurea or aphidicolin resulted in significantly stronger melanoma growth inhibition in cell culture (*Figure 9C*) and in mice (*Figure 9D*) than either drug alone. These results were surprising because *IFI6* was also upregulated by the BRAFV600E mutation. Therefore, we asked if *E2F2* and its target genes involved in the regulation of DNA replication were upregulated in BRAF-mutant, NF1-deficient, or triple wild-type melanoma cells deficient in *IFI6*. To do so, we knocked down the expression of *IFI6* and measured the expression of *E2F2* and its target genes (*Figure 9—figure supplement 1*). We did not observe similar changes in mRNA levels of *E2F2* or its target genes after *IFI6* knockdown in BRAF-mutant or NF1-deficient melanoma cell lines (*Figure 9—figure supplement 1*). These results indicate that only in the context of NRASQ61K, *IFI6* loss results in *E2F2* upregulation, which induces the expression of E2F2 target genes that regulate DNA replication. Thus, the ability of IFI6 to regulate DNA replication stress originates in its ability to repress *E2F2* and its target genes specifically in the context of NRAS-mutant melanoma. Collectively, these results demonstrate that NRASQ61K upregulates *IFI6*, consequently repressing *E2F2*, thereby conferring resistance to drugs that induce cytotoxicity by causing DNA replication stress. This genetically vulnerable pathway can be targeted pharmacologically by combining a MEK inhibitor with cytotoxic agents that induce DNA replication stress to potently inhibit NRAS-mutant melanoma growth.

## Discussion

In this report, we show that IFI6 regulates oncogenic NRAS-induced melanocyte transformation and NRAS-mutant melanoma tumor growth. Our findings are summarized in *Figure 10* and described below. First, our results demonstrate that IFI6 expression is activated by oncogenic NRAS, which is necessary for oncogenic NRAS-induced transformation and melanoma tumor growth. Second, we report a previously undocumented role for IFI6 in E2F2-mediated regulation of DNA replication. We show that oncogenic NRAS stimulates IFI6 expression to facilitate melanocyte transformation and tumor growth. The loss of IFI6 results in E2F2-mediated dysregulation of DNA replication, resulting in cellular senescence or apoptosis and tumor growth inhibition. Finally, we show that NRAS-mutant

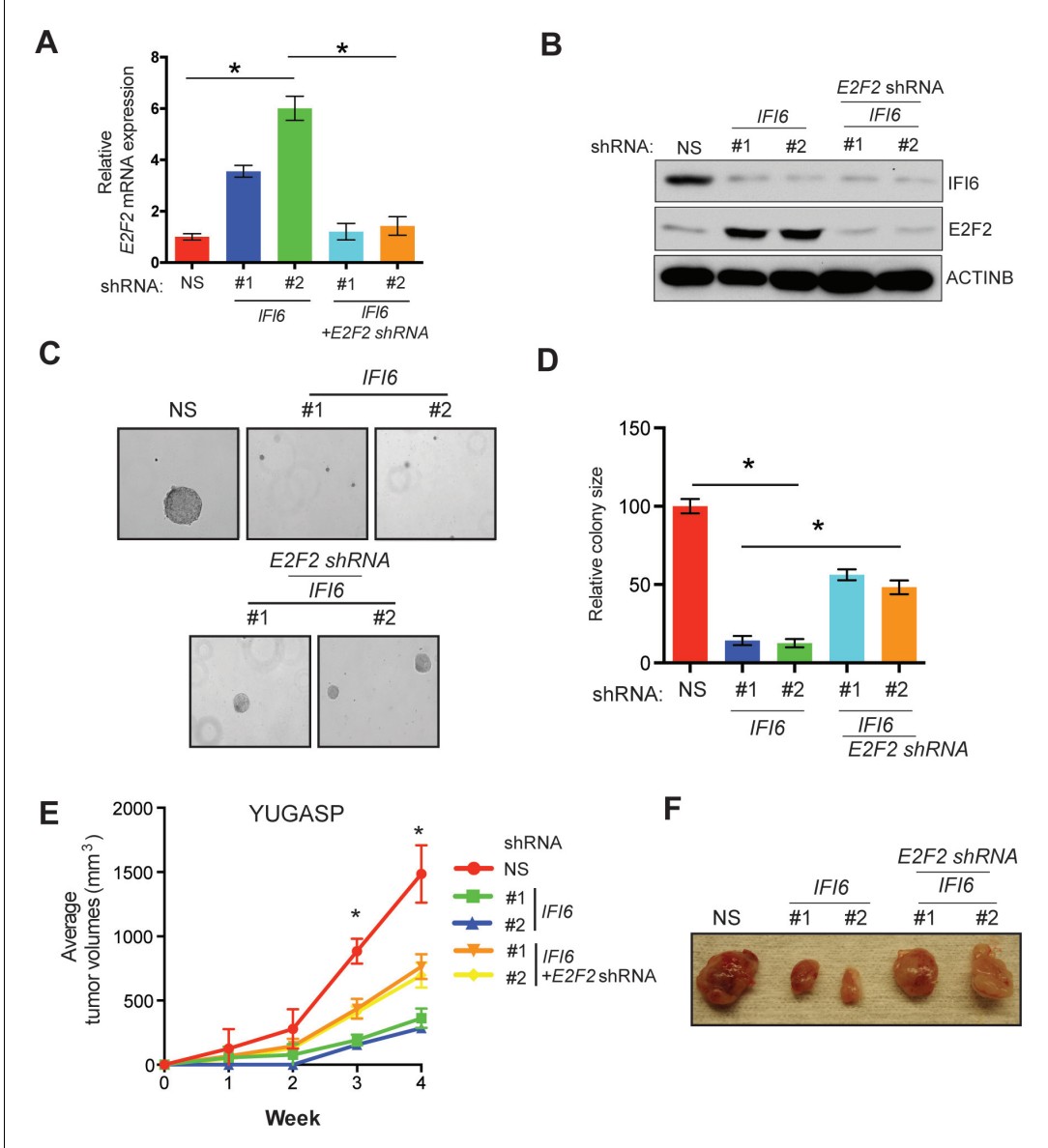

**Figure 5.** E2F2 mediates the loss of *IFI6*-induced tumor suppression. (**A**) YUGASP cells expressing either non-specific (NS) or *IFI6* shRNA, alone or in combination with *E2F2* shRNA, were analyzed for *E2F2* mRNA expression by RT-qPCR. A relative *E2F2* mRNA expression under the indicated conditions is shown. (**B**) YUGASP cells expressing either NS or *IFI6* shRNA, alone or in combination with *E2F2* shRNA were analyzed for E2F2, IFI6, and ACTINB protein expression by immunoblotting. (**C**) YUGASP cells expressing either NS or *IFI6* shRNA, alone or in combination with *E2F2* shRNA were analyzed for colony-forming potential. Representative soft agar images are shown. (**D**) Relative colony size in the soft agar assay presented in panel C under the indicated conditions is shown. (**E**) YUGASP cells expressing either NS or *IFI6* shRNA, alone or in combination with *E2F2* shRNA, were injected subcutaneously into the flank of athymic nude mice. Average tumor volumes (n = 5) formed from each cell line at the indicated times are shown. (**F**) Representative tumor images for the experiment presented in panel E under the indicated conditions are shown. In all panels, data are presented as mean ± SEM, and *p<0.05.

melanomas are significantly more resistant to cytotoxic agents that function by inducing DNA replication stress than wild-type NRAS melanoma cells. These observations are highly significant because they uncover a previously undocumented genetic vulnerability that is pharmacologically amenable and can be targeted to treat NRAS-mutant melanoma. These results also have important clinical implications because there are currently no effective therapies for NRAS-mutant melanoma.

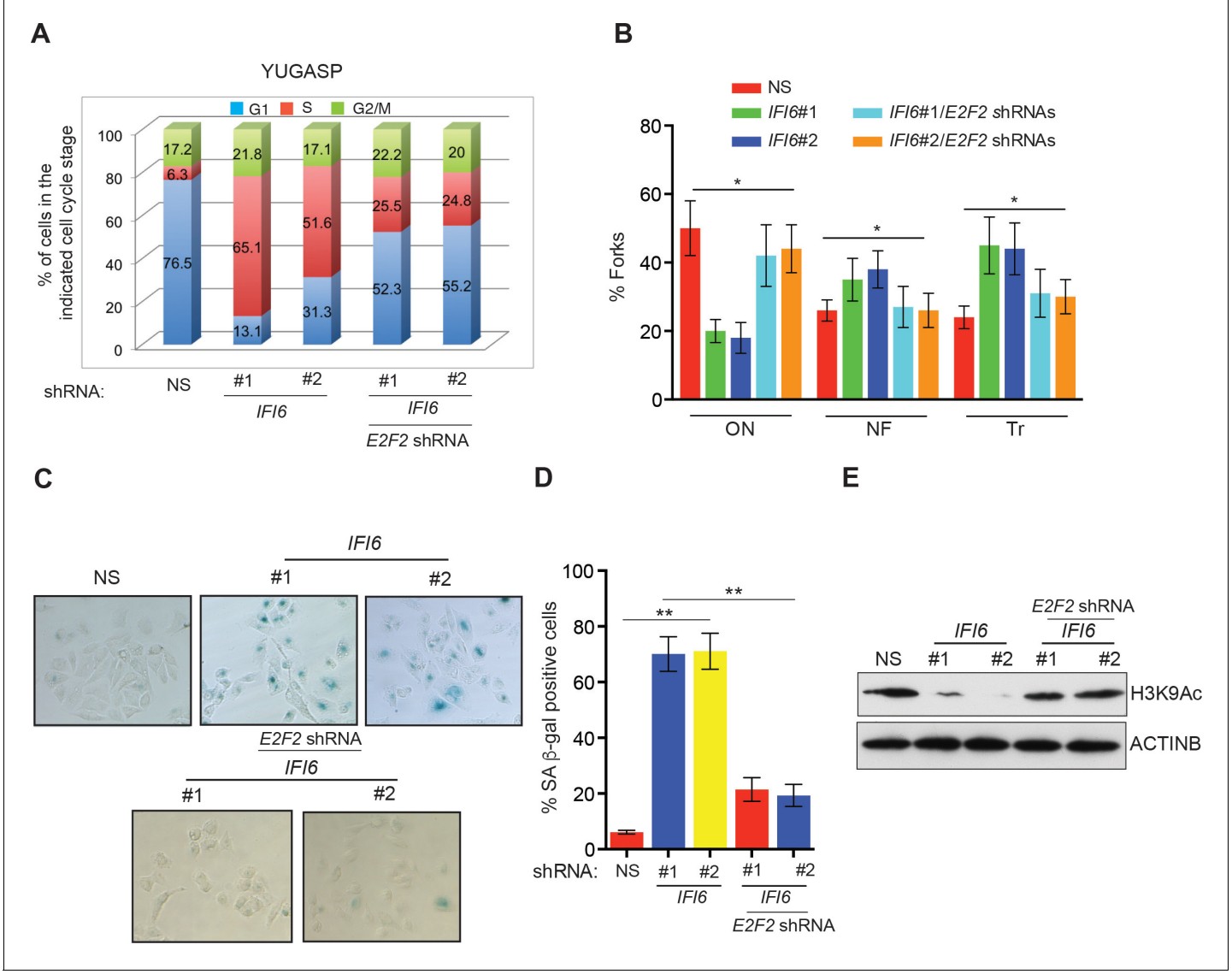

**Figure 6.** E2F2 mediates the loss of *IFI6*-induced dysregulated DNA replication. (**A**) YUGASP cells expressing either non-specific (NS) or *IFI6* shRNA, alone or in combination with *E2F2* shRNA were analyzed by FACS. The percentages of cells in each cell cycle stage are shown. (**B**) YUGASP cells expressing either NS or *IFI6* shRNA, alone or in combination with *E2F2* shRNA, were analyzed using the DNA fiber assay. The percentages of ongoing (ON), newly fired (NF), and terminated (Tr) DNA forks are shown. (**C**) YUGASP cells expressing either NS or *IFI6* shRNA, alone or in combination with *E2F2* shRNA, were analyzed for SA-β-gal activity. Representative images of cells stained for SA-β-gal activity are shown. (**D**) Percentage of SA-β-gal-positive cells for the experiment presented in the panel **C** is plotted. (**E**) YUGASP cells expressing each shRNA were analyzed for H3K9Ac by immunoblotting. ACTINB was used as a loading control. In all panels, data are presented as mean ± SEM, and *p<0.05, **p<0.005.

The following figure supplement is available for figure 6:

**Figure supplement 1.** E2F2 shRNA does not affect the mRNA expression of other E2F family genes.

## IFI6 is necessary for oncogenic NRAS-induced transformation and melanoma growth

ISGs are shown to play several important biological roles. Among these, their roles in resisting and controlling pathogens are well established (*Schneider et al., 2014*). For example, many RNA viruses that induce the interferon response activate the expression of ISGs. This activation allows host cells to control the viral infection (*Schneider et al., 2014*). This phenomenon is observed in patients

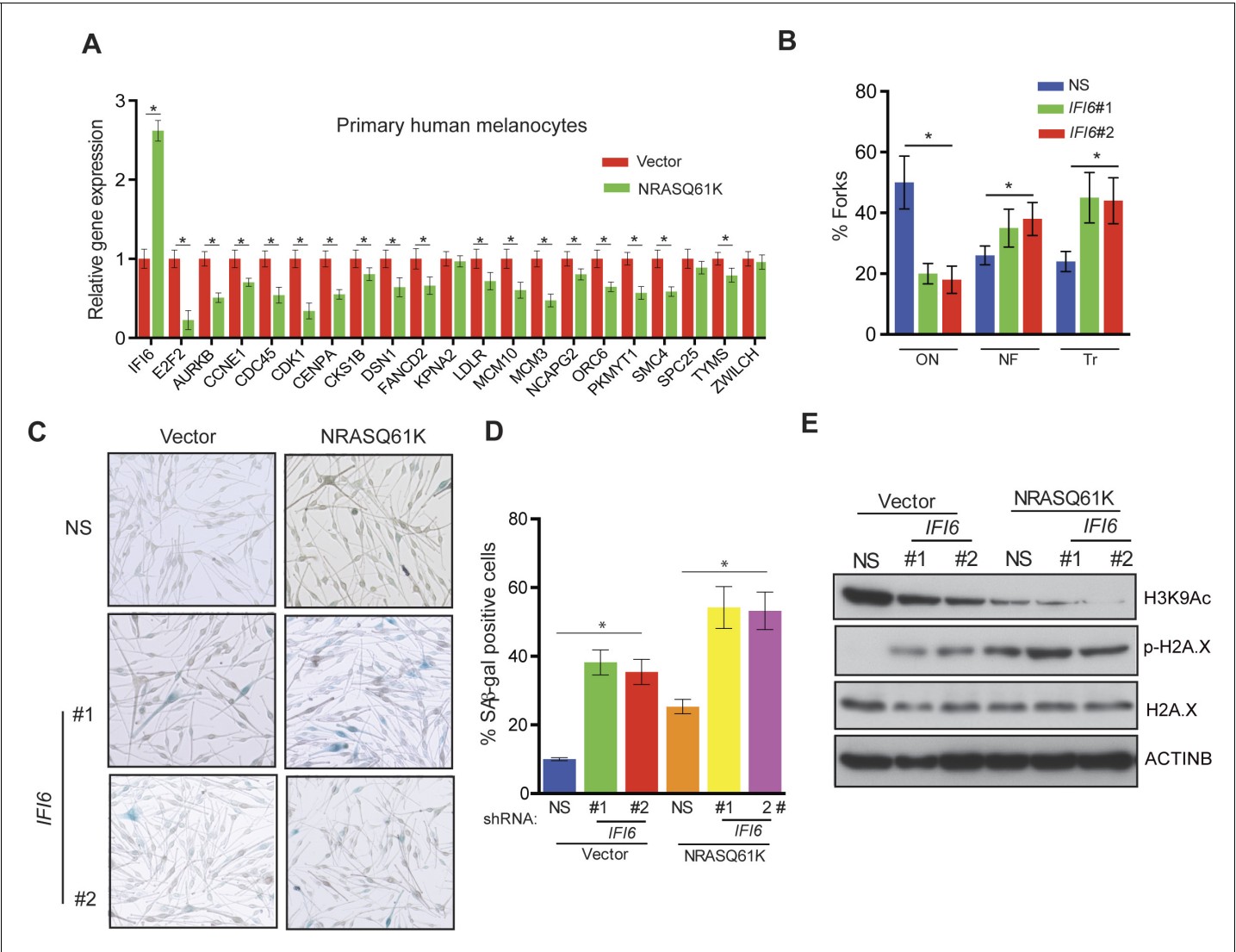

**Figure 7.** *IFI6* loss results in DNA replication stress and senescence induction in primary human melanocytes. (**A**) Primary human melanocytes expressing NRASQ61K or an empty vector were analyzed for the indicated genes by qRT-PCR. Relative gene expression is shown. (**B**) Primary human melanocytes expressing indicated shRNA were analyzed by DNA combing assay. Percentages of ongoing (ON), newly fired (NF), and terminated (Tr) DNA forks are shown. (**C**) Melanocytes expressing non-specific (NS) or *IFI6*, shRNA with either empty vector or NRASQ61K, were analyzed by SA-β-gal assay. Representative images are shown. (**D**) Percentage of SA-β-gal–positive cells expressing NS or *IFI6* shRNA with either vector or NRASQ61K for the experiment presented in panel **C** is shown. (**E**) Melanocytes expressing NS or *IFI6* shRNA in combination with either empty vector or NRASQ61K, were analyzed for the indicated proteins by immunoblotting. Data are presented as mean ± SEM, and *p<0.05.

The following figure supplement is available for figure 7:

**Figure supplement 1.** DNA fiber assay results of primary human melanocytes expressing *IFI6* shRNA.

infected with hepatitis C virus (HCV) who receive IFN-α-based drug regimens. In patients who respond to this treatment, expression of ISGs is low in the liver before treatment and increases significantly after treatment (*Sarasin-Filipowicz et al., 2008*). Furthermore, increased expression of 36 unique ISGs correlated with a reduction in HCV viral load (*Brodsky et al., 2007*). It is important to note that some viruses, such as human cytomegalovirus (HCMV), have been shown to use specific ISGs, such as RSAD2, to enhance infection (*Seo et al., 2011*). We found that three ISGs—IFI6, IFI27, and MX1—were transcriptionally upregulated by oncogenic NRAS and other oncogenic RAS proteins in a MAPK pathway-dependent manner. These results show that the introduction of oncogenic

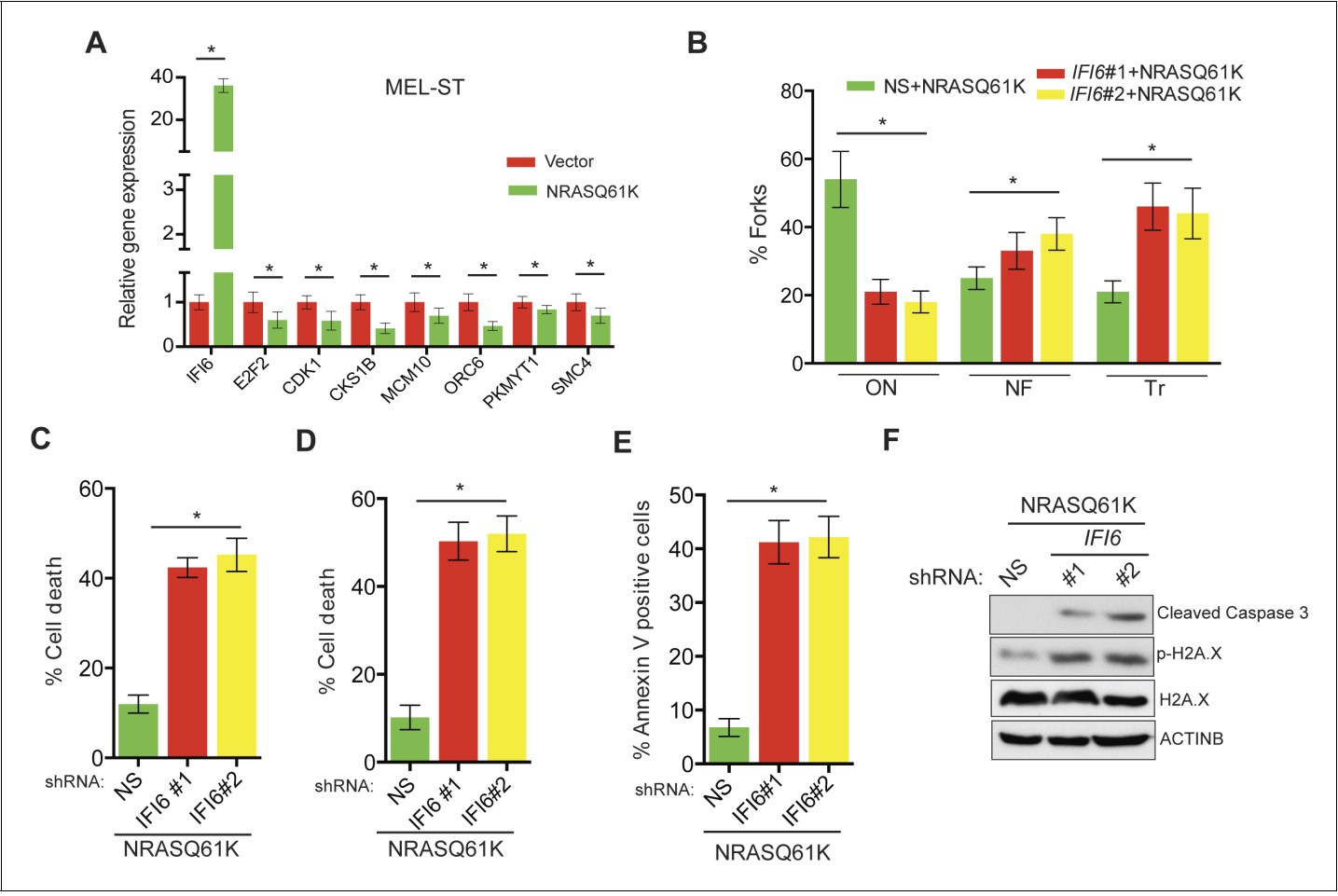

**Figure 8.** *IFI6* loss results in DNA replication stress and apoptosis induction in MEL-ST/NRASQ61K cells. (**A**) MEL-ST cells expressing empty vector or NRASQ61K were analyzed by RT-qPCR. The expression of each gene in MEL-ST/NRASQ61K cells is shown relative to expression in empty vector control. (**B**) MEL-ST/NRASQ61K cells with the indicated shRNA were analyzed for DNA replication using the DNA fiber assay. The percentages of ongoing (ON), newly fired (NF), and terminated (Tr) DNA forks are shown. (**C**) Growth In Low Attachment (GILA) assay was performed using MEL-ST/NRASQ61K cells expressing either non-silencing (NS) or *IFI6* shRNA. Cell death (%) was measured 48 hr after plating under the indicated conditions using the trypan blue exclusion assay and plotted. (**D**) MEL-ST/NRASQ61K cells expressing either NS or *IFI6* shRNA were plated on poly-HEMA plates. Cell death (%) was measured 48 hr after plating under each condition using the trypan blue exclusion assay and plotted. (**E**) MEL-ST/NRASQ61K cells expressing either NS or *IFI6* shRNA were plated on poly-HEMA plates. Apoptotic cell death (%) was measured 48 hr after plating under each condition by annexin V-FITC staining and plotted. (**F**) MEL-ST/NRASQ61K cells expressing either NS or *IFI6* shRNA were plated on poly-HEMA plates. After 48 hr, the indicated proteins were analyzed by immunoblotting. ACTINB was used as a control. In all panels, data are presented as mean ± SEM, and *p<0.05.

The following figure supplements are available for figure 8:

**Figure supplement 1.** DNA fiber assay results in MEL-ST/NRASQ61K cells.

**Figure supplement 2.** *IFI6* knockdown in MEL-ST/NRASQ61K does not induce senescence.

RAS into human cells induces responses similar to those seen during viral infections, and oncogenic NRAS uses ISGs, such as IFI6, to facilitate cellular transformation and tumor growth.

## Loss of IFI6 results in E2F2-mediated DNA replication stress

In addition to identifying the role for IFI6 in the regulation of melanoma development and tumor growth, we also discovered a previously undocumented role for IFI6 in the regulation of DNA replication. We found that the inhibition of *IFI6* by shRNA-mediated knockdown results in upregulation

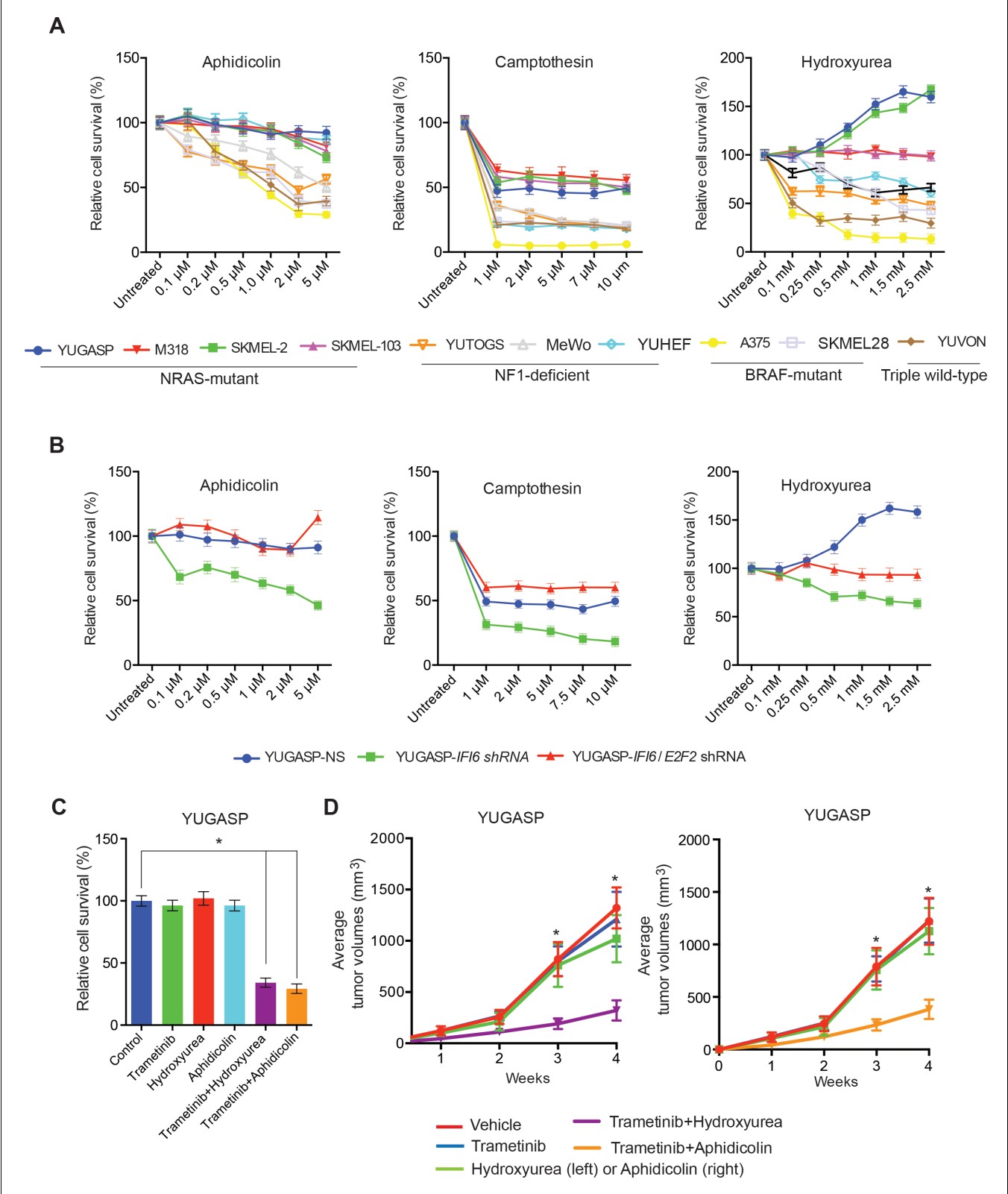

**Figure 9.** Targeting the DNA replication stress resistance pathway to treat NRAS-mutant melanoma. (**A**) Indicated melanoma cell lines were treated with indicated drugs and analyzed by MTT assay after 48 hr of drug treatment. Survival was determined relative to vehicle-treated cells. (**B**) Indicated NRAS-mutant melanoma cell lines expressing *IFI6* or NS shRNA, or simultaneously expressing *IFI6* and *E2F2* shRNA, were treated with each DNA replication stress-inducing agent for 48 hr and analyzed by MTT assay. Survival was determined relative to cells expressing NS shRNA. (**C**) Melanoma

*Figure 9 continued on next page*

*Figure 9 continued*

cells were treated with trametinib (1 nM) alone or in combination with DNA replication stress-inducing agents (0.2 μM aphidicolin or 0.25 mM hydroxyurea) for 48 hr and analyzed by MTT assay. Survival was determined relative to vehicle-treated cells. (**D**) Melanoma cells were injected subcutaneously into athymic nude mice (n = 5). The mice were treated on alternate days with trametinib (0.1 mg/kg, orally) alone or in combination with DNA replication stress-inducing agents (50 mg/kg aphidicolin i.p. or 50 mg/kg hydroxyurea, i.p.). Average volumes for tumors formed from each cell line at the indicated times are shown (n = 5). In all panels, data are presented as mean ± SEM, and *p<0.05.

The following figure supplement is available for figure 9:

**Figure supplement 1.** *IFI6* knockdown in non-NRAS mutant melanoma cells does not induce expression of *E2F2* or its target genes.

of transcription factor E2F2, which specifically activates its target genes involved in the activation of DNA replication. This upregulation of DNA replication-promoting genes results in dysregulated

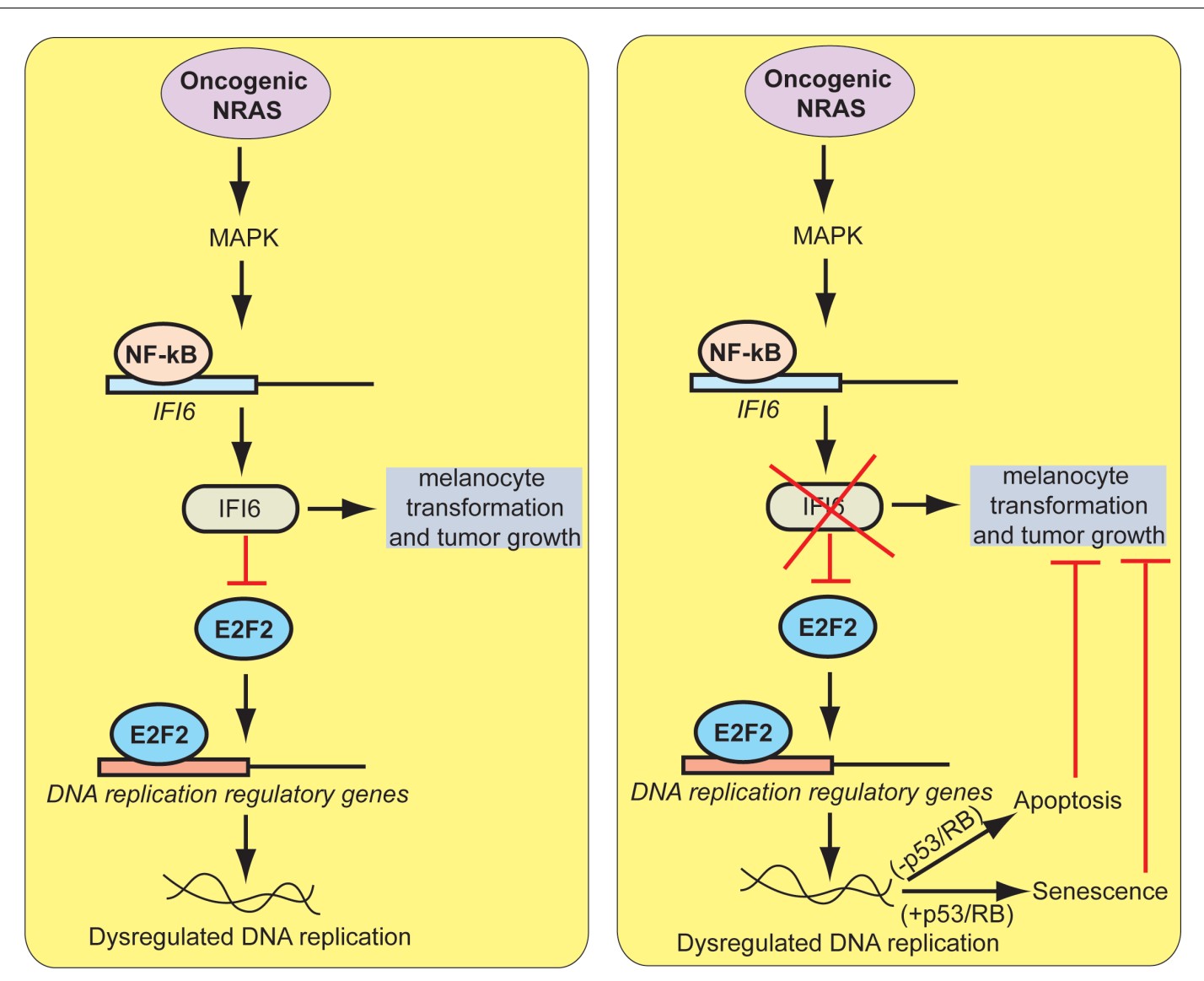

**Figure 10.** Proposed model for the role of IFI6 in melanoma tumor growth. The model shows the mechanism by which IFI6 contributes to NRASQ61K-induced transformation and the maintenance of oncogenic NRAS-mutant melanoma tumor growth.

DNA replication in melanocytes and melanoma cells. Some studies have shown that oncogenes induce DNA replication stress, resulting in cellular senescence (*Bartkova et al., 2006*; *Di Micco et al., 2006*), but the precise mechanism of this process was not clearly illustrated. Here, we provide evidence that loss of IFI6 results in the upregulation of *E2F2*, which dysregulates DNA replication. This biological process is also necessary for NRAS-induced cellular transformation and for maintaining the NRAS-mutant melanoma tumor growth. The E2F transcription factors are downstream effectors of the retinoblastoma pathway, and their transcriptional activity is required to regulate various genes, including those required for DNA replication and cell cycle progression (*Bracken et al., 2004*). Context-specific roles for different E2F family members have been shown, which further highlights their non-redundant biological functions (*Attwooll et al., 2004*). Interestingly, we found that the loss of IFI6 results in specific upregulation of E2F2 but not other E2F family genes. This observation suggests a tumor suppressor-like role for E2F2 in NRAS-mutant melanoma. Notably, although E2F family proteins are typically known to activate genes that facilitate cell cycle progression and proliferation, a number of growth inhibitory genes, such as CDKN2C (p18), CDKN2D (p19), CDKN1C (p57), E2F7, RB1 (pRB), and RBL1 are upregulated by E2F proteins during S phase (*Di Stefano et al., 2003*; *Ortega et al., 2002*; *Sherr and Roberts, 1999*; *Stott et al., 1998*). Thus, one might speculate that the induction of these growth inhibitory genes might be necessary for regulated progression of the cell cycle. For example, activation of cyclin-dependent kinase inhibitors, such as CDKN2C, CDKN2D, and CDKN1C, would hypothetically reduce CDK activity and DNA replication fork firing during late S phase. Interestingly, none of these growth inhibitory targets of E2F were upregulated after *IFI6* knockdown, which may in part explain why IFI6-mediated upregulation of E2F2 results in dysregulated DNA replication and tumor growth inhibition.

## Cellular senescence or apoptosis as an outcome of loss of *IFI6* expression

Our study also shows that loss of *IFI6* results in two different outcomes, depending on the status of the p53 and RB pathways. In melanoma cells and primary human melanocytes, *IFI6* loss results in the induction of cellular senescence as a result of E2F2-mediated dysregulated DNA replication and consequent DNA damage. However, oncogenic NRAS-transformed MEL-ST cells express SV40 early region, thus inactivating both p53 and RB pathways. In this scenario, we found that the loss of *IFI6* results in apoptosis induction. Our results are consistent with multiple previous studies that have demonstrated DNA damage in p53 and RB pathway-defective cells, resulting in p53- and RB-independent apoptosis (*Aladjem et al., 1998*; *Knudsen et al., 2000*; *Strasser et al., 1994*).

## DNA replication stress resistance as a genetic vulnerability of NRAS-mutant melanoma

Our studies have identified a novel and clinically significant feature of NRAS-mutant melanoma. We found that cytotoxic agents that typically inhibit growth by inducing DNA replication stress are largely ineffective at inhibiting NRAS-mutant melanomas. In contrast, BRAF-mutant, NF1-deficient, and triple wild-type melanomas were relatively sensitive to these agents. We also confirm resistance to these agents is due in part to the oncogenic NRAS-mediated upregulation of IFI6 and consequential repression of E2F2 and its target genes. Knockdown of *IFI6* increased the sensitivity of NRAS-mutant melanoma to DNA replication stress-inducing agents, and simultaneous loss of E2F2 and IFI6 rescues this effect. Notably, knockdown of *IFI6* in BRAF-mutant, NF1-deficient, or triple wild-type melanoma did not alter the expression of *E2F2* or its target genes. This in part explains the differences in the sensitivity of melanoma genotypes to agents that induce cytotoxicity through DNA replication stress. Because IFI6 is upregulated in NRAS-mutant melanoma, and its expression correlates with NRAS mutation status, this subgroup of melanoma could be targeted using this therapeutic approach. Therefore, the DNA replication stress pathway represents a unique vulnerability of NRAS-mutant melanoma that can be pharmacologically inhibited to achieve tumor inhibition in cell culture and in vivo. Several of the agents we used here, such as aphidicolin, camptothecin, and hydroxyurea, are used in clinical cancer treatment (*Geyer and Mesa, 2014*; *Patel et al., 2006*; *Sargent et al., 1996*). Based on our results, we speculate that these drugs will have limited utility in patients harboring NRAS and potentially other RAS mutations. Thus, our results may have clinical application for the identification of patient populations that might benefit from these drugs.

# Materials and methods

## Cell culture and plasmids

SKMEL-2, SKMEL-28, MeWo, and A375 cells were purchased from American Type Culture Collection (ATCC) and grown as recommended. Neonatal primary human melanocytes were purchased from Life Technologies and grown as recommended by the supplier. All short-term melanoma cultures (YUGASP, YUHEF, YUTOGS, and YUVON) were obtained from the Yale SPORE in Skin Cancer, Yale University, and grown as recommended. SKMEL-103 and M318 cells were provided by Dr. Keiran Smally (Moffitt Cancer Center, Florida). MEL-ST cells were provided by Prof. Robert Weinberg (Whitehead Institute, MIT). All the cell lines were authenticated using STR analysis and tested for mycoplasma regularly using a MycoAlert Mycoplasma detection kit (Lonza, Allendale, NJ). Human *IFI6* open reading frame was cloned in pcDNA3.1/hygro and used for rescue experiments. The plasmids pBabe puro-HRAS V12 (plasmid #15269), pBabe puro-HRAS V12 S35 (plasmid# 12274), pBabe puro-MEK-DD (plasmid #15268), and pBabe-puro-BRAFV600E (plasmid #15269) were purchased from Addgene. FG12 was a kind gift of Prof. David Baltimore, and FG12/NRASQ61K was a kind gift of Maria Soengas (CNIO, Spain).

## Microarray experiments and data analysis

For microarray experiments using MEL-ST cells, total RNA was isolated from MEL-ST cells transduced by FG12-NRASQ61K or vector control and used to generate labeled antisense RNA. For microarray experiments using YUGASP cells, total RNA was isolated from YUGASP cells expressing either non-specific (NS) or one of two different *IFI6* shRNA sequences and used to generate labeled antisense RNA. All antisense RNAs were made using the Ambion MessageAmp Kit and hybridized to Illumina HumanHT-12 V4.0 expression BeadChip using Illumina's protocol.

The microarray data were processed using GenomeStudio (Illumina), log2-transformed and quantile-normalized using the 'lumi' package of Bioconductor. All samples passed quality-control (QC) assessment, which included checking various control plots as suggested by Illumina, as well as other standard microarray-related analyses. Differential expression analyses were performed using the 'limma' package, and a moderated $t$-test with a Benjamini-Hochberg multiple testing correction procedure was used to determine statistical significance (adjusted p-value, <0.05). Pathway analysis of differentially expressed genes for each comparison was performed using MetaCore (version 6.8 build 29806; GeneGo). Microarray data were submitted to Gene Expression Omnibus (Accession No. GSE39294 for MEL-ST cell experiments, and GSE69933 for YUGASP experiments).

## Melanoma dataset analysis

The Talantov melanoma dataset (*Talantov et al., 2005*), Haqq melanoma dataset (*Haqq et al., 2005*), Barretina cell line dataset (*Barretina et al., 2012*), and Riker melanoma dataset (*Riker et al., 2008*) were downloaded from Oncomine (https://www.oncomine.org) and analyzed for IFI6 expression and plotted as box plots between melanoma and normal skin samples. The Talantov melanoma dataset analyzed 45 cutaneous melanomas, 18 benign melanocytic skin nevi, and 7 normal skin samples using Affymetrix U133A microarrays. The Haqq melanoma dataset analyzed 25 melanomas, 9 non-neoplastic nevi, and 3 normal skin samples using a cDNA microarray. The Cancer Cell Line Encyclopedia (CCLE) project (i.e., Barretina cell line dataset) is a compilation of gene expression data from 917 human cancer cell lines comprising 18 different tumor types. Lane 13 represents 57 melanoma cell lines. The Riker melanoma dataset analyzed 40 metastatic melanomas, 42 primary skin cancers, 4 normal skin samples, and 1 normal skin primary cell culture using Affymetrix HG U133 Plus 2.0 microarrays.

## RNA preparation, cDNA preparation, and quantitative PCR analysis

For mRNA expression analyses, total RNA was extracted with TRIzol (Invitrogen) and purified using RNeasy mini columns (Qiagen). The cDNA was generated using the M-MuLV first-strand cDNA synthesis kit (New England Biolabs) according to the manufacturer's instructions. Quantitative RT-PCR was performed using the Power SYBR Green kit (Applied Biosystems) according to the manufacturer's instructions. Actin was used as an internal control. Primer sequences are provided in *Supplementary file 1D*.

## Chromatin immunoprecipitation

The *IFI6* promoter sequence was downloaded from the UCSC genome browser and analyzed using rVista 2.0. ChIP experiments were performed as described previously (*Gazin et al., 2007*). Normalized Ct ($\Delta$Ct) values were calculated by subtracting the Ct obtained with input DNA from that obtained with immunoprecipitated DNA [$\Delta$Ct = Ct(IP) − Ct(input)]. Relative fold enrichment of a factor at the target site was then calculated using the formula $2^{-[\Delta Ct(T) - \Delta Ct(Actb)]}$, where $\Delta$Ct(T) and $\Delta$Ct (Actb) are $\Delta$Ct values obtained using target and *Actb* (irrelevant) primers, respectively.

## shRNA, retrovirus, and lentivirus preparation

*IFI6, E2F2, STAT1*, and *RelA* (the p65 subunit of NF-$\kappa$B), *NF1* shRNA sequences in either pLKO.1 or pZIPZ lentiviral expression vectors were obtained from Open Biosystems. For retrovirus or lentivirus production viral expression, constructs, and viral packaging plasmids were co-transfected into 293T cells using Effectene (Qiagen), following the manufacturer's recommendations. Supernatant fractions were collected 48 hr after the transfection, and purified virus particles were used to infect primary or melanoma cell lines and were selected by growth on puromycin or by sorting GFP-positive cells using a flow cytometer.

## Antibodies and immunoblot analysis

Whole cell protein extracts were prepared using IP lysis buffer (Pierce) containing Protease Inhibitor Cocktail (Roche) and Phosphatase Inhibitor Cocktail (Sigma-Aldrich, St. Louis, MO). Protein concentration was estimated using a Bradford Assay kit (Bio-Rad). Proteins were resolved on 10% or 12% polyacrylamide gels and transferred to PVDF membranes using a wet transfer apparatus from Bio-Rad. Membranes were blocked with 5% skim milk and probed with primary antibodies, followed by the appropriate secondary HRP-conjugated antibody (GE Healthcare, UK). Blots were developed using the SuperSignal Pico Reagent (Pierce). Information about the antibodies used in this study is provided in *Supplementary file 1D*.

## Melanoma sample analysis

Total RNA from melanoma samples were obtained from the University of Massachusetts Medical School (UMMS) Tissue and Tumor Bank and analyzed for the expression of *IFI6* and MAPK target genes *FOSL1, ETV5, SPRY2*, and *DUSP6* by RT-qPCR.

## Soft agar assay and tumorigenesis assay

For the soft agar assay, individual cell lines were seeded in triplicate at three different dilutions, ranging from $5 \times 10^3$ to $2 \times 10^4$ cells. Cells were seeded into a 0.4% soft agar layer. After 4 weeks, colonies were stained with 0.005% crystal violet solution and counted. Each experiment was repeated at least twice. Athymic nude (NCr nu/nu) mice (aged six weeks) were injected subcutaneously with cell lines either expressing different shRNAs or transduced with the empty vector. Tumor volume was calculated using the formula: length $\times$ width$^2$ $\times$ 0.5. All animal protocols were approved by the Institutional Animal Care and Use Committee (IACUC) at Yale University. For mouse experiments trametinib was dissolved in 0.5% hydroxypropyl methylcellulose and was administered orally (0.1 mg/kg) every other day. Hydroxyurea and aphidicolin were dissolved in sesame oil and administered by intraperitoneal (i.p.) injection every other day (50 mg/kg) alone or in combination with trametinib (0.1 mg/kg).

## DNA fiber assay

The DNA fiber assay was performed as described previously (*Merrick et al., 2004*). Briefly, cells were plated in the appropriate medium until they reached 30–40% confluency. After 48 hr, IdU (Sigma-Aldrich: I7125) was added to the exponentially growing cells (final concentration: 25 $\mu$M), and the cells were incubated for 30 min at 37°C in 5% $CO_2$. After washing with PBS, cells were incubated with a second label, CldU (Sigma-Aldrich: C6891), at the final concentration of 250 $\mu$M for additional 30 min at 37°C. Cells were trypsinized and counted, and $2 \times 10^3$ cells in 3 $\mu$l of medium were used for each slide. The 3 $\mu$l of cell suspension was applied to the end of the glass slide and air-dried for 5 min. Cells were lysed by adding 7 $\mu$l of lysis solution (50 mM EDTA and 0.5% SDA in 200 mM Tris-HCl, pH 7.6). Glass slides were placed at a 15° angle to allow the DNA fibers to spread

across the length of the slide, and then were placed horizontally to air dry. After this, slides were fixed with methanol:acetic acid (3:1) for 10 min, washed with double distilled water, and treated with 2.5 M HCl for 30 min. Next, the fixed cells were blocked with 5% BSA for 30 min at room temperature and incubated with primary antibodies (anti-BrdU [mouse antibody], BD Biosciences #347580 for IdU at a 1:25 dilution and anti-BrdU [rat antibody], Abcam # ab6326 for CldU at a 1:400 dilution, each in 5% BSA) for 1 hr at room temperature in a humidified chamber. Slides were then washed three times with $1\times$ PBS for 5 min and then incubated with secondary antibodies (1:500 sheep anti-mouse Cy3, Sigma, Cat# C218-M for IdU, and 1:400 goat anti-rat Alexa Fluor 488, Invitrogen, cat A11006 for CldU) in 5% BSA for 1 hr at room temperature in the dark. After secondary antibody incubation, glass slides were washed and visualized at 63X magnification to locate the fibers. Pictures were captured with one color channel, and data were analyzed with image analysis software ImageJ.

## Senescence-induced beta-galactosidase (SA-β-gal) staining

The SA-β-gal assay was performed as described previously (*Dimri et al., 1995*). Briefly, $2.0 \times 10^5$ cells were plated in 6-well plates and stained after 48 hr. First, the cells were washed twice with 1X PBS and fixed with 4% paraformaldehyde for 5 min at 37°C. Then the cells were washed twice more with 1X PBS and incubated with β-gal staining solution (made fresh in $1\times$ PBS: X-gal 1 mg/ml (0.1%), potassium ferricyanide 5 mM, potassium ferrocyanide 5 mM, $MgCl_2$ 2 mM, NaCl 150 mM, and citric acid/sodium phosphate solution 40 mM) in a 37°C incubator in the dark. The reaction was terminated by removing the staining solution and washing the cells twice with distilled water. Cells were visualized with an inverted bright-field microscope, and the images were captured using the $10\times$ objective. The percentage of SA-β-gal–positive cells was plotted with respect to the total number of cells used in each case.

## Fluorescence-activated cell sorting (FACS)

FACS analyses were conducted as described previously (*Santra et al., 2009*). Briefly, cells were fixed with 70% ethanol overnight. The following day, cells were washed twice with $1\times$ PBS and suspended in 300 µl of $1\times$ PBS, treated with RNase (Sigma-Aldrich) and propidium iodide for 1 hr, and analyzed using FACSCalibur (BD Biosciences).

## MTT assay

For this assay, $5 \times 10^3$ cells were plated in a 100 µl volume in 96-well plates. After 48 hr, inhibitors (i. e., aphidicolin, camptothecin, hydroxyurea, and trametinib), used at a range of concentrations was mixed in 100 µl of medium and added to the cells. After 48 hr of inhibitor treatment, the cell viability was evaluated. To do this, 20 µl of 5 mg/ml MTT solution dissolved in $1\times$ PBS was added to each well and incubated for 1 hr at 37°C incubator. The MTT solution was removed gently, and 100 µl of DMSO were added. After mixing well by pipetting, absorbance was measured at 590 and 630 nm. An average was calculated for both readings, and then measurement at 630 nm was subtracted from that at 590 nm. The relative growth rate was plotted with respect to vehicle-treated control cells.

## Luciferase reporter assay

MEL-ST/NRASQ61K cells expressing *IFI6* shRNA were transfected with a NF-κB-firefly luciferase construct (Promega) along with a control TK-Renilla luciferase plasmid (pRL-TK). After 24 hr, the cells were lysed in passive lysis buffer, and the luciferase reporter assay was performed using the Dual-Luciferase Reporter Assay Kit (Promega). Relative NF-κB activity was measured as the ratio of firefly to Renilla luciferase activity and reported as the average of triplicate measurements.

## Poly-HEMA assay and growth in low attachment (GILA) assay

To evaluate anoikis, 6-well tissue culture plates were coated with 200 µL poly-HEMA (Sigma) and left for 10 hr in a laminar flow hood. MEL-ST/NRASQ61K cells expressing NS or *IFI6* shRNA were seeded in the poly-HEMA-coated plates and incubated for 48 hr at 37°C. The cells were then counted using trypan-blue exclusion assay by a hemocytometer. The experiments were performed in triplicate.

GILA assay was performed as described previously (*Rotem et al., 2015*). Briefly, 25,000 MEL-ST/ NRASQ61K cells expressing NS or *IFI6* shRNA were seeded in 12-well ultra low-attachment plates (Corning) and incubated for 48 hr at 37°C. The cells were then counted using trypan-blue exclusion assay by a hemocytometer. The experiments were performed in triplicate.

## Apoptosis measurement using annexin V/ propidium iodide staining and cleaved caspase 3 immunoblot

MEL-ST/NRASQ61K cells expressing NS or *IFI6* shRNA were analyzed for apoptosis by flow cytometry using the FITC-Annexin Apoptosis Detection Kit I (BD Pharmingen), as per the manufacturer's protocol. Caspase 3 cleavage was detected by immunoblotting using an anti-cleaved caspase 3 antibody (Cell Signaling Technology).

## Statistical analysis

All the experiments were conducted in three biological replicates. The results for individual experiments were expressed as mean ± SEM. The *P*-values were calculated by *t*-test using GraphPad Prism version 6.0 hr for Macintosh, GraphPad Software, San Diego, California, USA (www.graphpad.com).

## Acknowledgements

We thank Prof. Joann Sweasy, Yale University, for help with the DNA combing assay. We gratefully acknowledge grants from the National Institutes of Health: R01CA200919 (NW), R01CA196566 (NW), R21CA197758 (NW), R21CA191364 (NW), R21CA195077 (NW), and NW is also supported by a Research Scholar Grant from the American Cancer Society (128347-RSG-15-212-01-TBG) and grants from the Melanoma Research Alliance.

## Additional information

### Funding

| Funder | Grant reference number | Author |
|---|---|---|
| National Institutes of Health | R01CA200919 | Narendra Wajapeyee |
| National Institutes of Health | R21CA195077-01A1 | Narendra Wajapeyee |
| National Institutes of Health | R21CA191364-01 | Narendra Wajapeyee |
| National Institutes of Health | R21CA197758-01 | Narendra Wajapeyee |
| Melanoma Research Alliance | Pilot grant award | Narendra Wajapeyee |
| National Institutes of Health | R01CA196566 | Narendra Wajapeyee |
| National Institutes of Health | R21CA197758 | Narendra Wajapeyee |
| American Cancer Society | 128347-RSG-15-212-01-TBG | Narendra Wajapeyee |
| Melanoma Research Alliance | | Narendra Wajapeyee |

The funders had no role in study design, data collection and interpretation, or the decision to submit the work for publication.

### Author contributions

RG, NW, Conception and design, Acquisition of data, Analysis and interpretation of data, Drafting or revising the article; MF, Conception and design, Acquisition of data, Analysis and interpretation of data; MB, Conception and design, Acquisition of data, Drafting or revising the article; SKD, Performed the analysis and interpretation of all microarray data presented in this manuscript and worked for GEO submission of the files; QY, Helped with the acquisition of data presented in the Figure 2 and Figure 3-figure supplements

### Author ORCIDs

Narendra Wajapeyee, http://orcid.org/0000-0003-3306-349X

## Ethics

Animal experimentation: This study was performed in strict accordance with the recommendations in the Guide for the Care and Use of Laboratory Animals of the National Institutes of Health. All of the animals were handled according to approved institutional animal care and use committee (IACUC) protocols of the Yale University (IACUC protocol #2016-11333).

# Additional files

### Supplementary files

• Supplementary file 1. Files related to microarray analysis and tables for reagents. (A) List of genes that are significantly upregulated ($p<0.05$, Fold-change = 2 fold or more) in NRASQ61K expressing MEL-ST cells. (B) Fold change (FC) for significantly altered genes (p-value<0.05) in YUGASP cells expressing IFI6 shRNAs. (C) Biological pathway enrichment analysis report. (D) Primer sequences for RT-qPCR analysis; clone ID and catalog numbers for shRNAs (Open Biosystems); antibodies used; source and concentration of chemical inhibitors used.

• Supplementary file 2. Analysis of MAP kinase regulated and BRAF-signature genes for correlation with BRAF/NRAS/NF1 mutation status using melanoma TCGA dataset.

### Major datasets

The following datasets were generated:

| Author(s) | Year | Dataset title | Dataset URL | Database, license, and accessibility information |
|---|---|---|---|---|
| Gupta R, Forloni M, Dogra S, Wajapeyee N | 2016 | Transcriptional targets of oncogenic RAS proteins that mediate their ability to induce cellular transformation | http://www.ncbi.nlm.nih.gov/geo/query/acc.cgi?acc=GSE62827 | Publicly available at the NCBI Gene Expression Omnibus (accession no: GSE62827). |
| Gupta R, Forloni M, Dogra S, Wajapeyee N | 2016 | Regulators of NRAS-mediated transformation and melanoma tumor maintenance | http://www.ncbi.nlm.nih.gov/geo/query/acc.cgi?acc=GSE69933 | Publicly available at the NCBI Gene Expression Omnibus (accession no: GSE69933). |

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
