## [Decision Letter]

Thank you for submitting your article "Interferon Αlpha-Inducible Protein 6 regulates NRASQ61K-induced melanomagenesis and growth by regulating DNA replication" for consideration by *eLife*. Your article has been reviewed by three peer reviewers, including Zhi Sheng (Reviewer #2) and a member of our Board of Reviewing Editors, and the evaluation has been overseen by Kevin Struhl as the Senior Editor.

The reviewers have discussed the reviews with one another and the Reviewing Editor has drafted this decision to help you prepare a revised submission. The required experiments represent a substantial investment and it will be good to hear back from you with a plan and an estimate of the time it may take to complete the required additions and changes.

The current manuscript by Gupta et al. describes the putative involvement of the interferon-stimulated gene IFI6 in the development of NRAS mutated melanoma. The authors describe a molecular pathway whereby mutations in the NRAS oncogene lead to upregulation of IFI6 expression via the activation of MEK kinases and the NF-κB transcription factor. Since IFI6 depletion leads to deregulated DNA replication and senescence, the authors propose a model whereby IFI6 upregulation downstream of NRAS is required for cell proliferation and tumor growth.

The reviewers and Senior Editor discussed the manuscript at length and after much deliberation reached a consensus, deciding to invite a resubmission of a thoroughly revised manuscript addressing the points below. A key point discussed by the reviewers is the fact that the mechanism illuminated is based on cell-based work and xenograft studies, and that a key test of the conclusions would involve the use of genetically-engineered mouse (GEM) models. However, the reviewers agreed that requiring the use of GEM models would be outside of the scope of the current manuscript, and that a thorough response to the concerns raised below could be satisfactory.

The major concerns are:

1) Specificity for NRAS-driven tumors. The manuscript describes a mechanism that is supposedly specific for NRAS-driven melanomas, but reviewers pointed out that the involvement of the RAF->MEK->ERK pathway indicates that melanomas driven by mutations in BRAF or silencing of NF1 should also display elevated IFI6 mRNA expression. This point should be carefully investigated and clarified. Is IFI6 upregulation unique to NRAS-driven tumors or not? Furthermore, by what mechanism(s) does NRAS-activated RAF->MEK->ERK activate NF-κB? Is it a direct mechanism of activation or does this occur through the secretion of inflammatory cytokines leading to NF-κB activation? If NRAS activates NF-κB signaling through the RAF->MEK->ERK pathway as described, then why wouldn't BRAF(V600E) or NF1(Null) melanomas, which have readily detectable pERK1/2 signaling, not also activate IFI6 mRNA expression? Indeed, it was noted that in their "NRAS wild-type" melanoma cell lines the authors employed at least one cell line (A375), which expresses BRAF(V600E) and has readily detectable pERK1/2 signaling. In sum, additional experiments are required to provide mechanistic insight into the mechanism of NRAS-mediated IFI6 expression through NF-κB activation and thus clarify the source of specificity (or lack thereof) for NRAS-driven melanoma.

2) Deregulated DNA replication versus senescence. The authors conclude that RNAi-mediated silencing of IFI6 leads to senescence, but this is not well documented and somewhat contradictory. The cells display elevated levels of SA- -β-gal, a widely used but unreliable marker of senescence, and a decrease in H3K9 acetylation. However, the cells also display increased numbers of cells in S phase, which is contrary to the idea of senescence, an irreversible form of cell cycle arrest generally associated with arrest in G1. How can these observations be reconciled? Could the authors detect evidence of DNA damage such as collapsed DNA replication forks, gH2AX foci and/or the recruitment of RAD51 or 53BP1 into damage sites?

Furthermore, in their initial gene induction studies and throughout the manuscript that authors employed Mel-ST cells, which are melanocytes immortalized by expression of hTERT and rendered susceptible to oncogenic transformation by expression of the SV40 Large and small T/t antigens. Since Large T binds to and inhibits the activity of the pRB, p107 and p130 "pocket proteins" that are negative regulators of E2F activity, and also to TP53, it is hard to understand mechanistically how senescence could be elicited in cells that are lacking these key tumor suppressors that are known to enforce the senescence program. Moreover, the authors claim that shIFI6 leads to elevated E2F2 mRNA expression (Figure 3) but since, to all intents and purposes, Mel-ST cells are TP53, pRB, p107 and p130 deficient, the significance of this observation is questionable since these cells should already have abundant E2F activity. It is also notable that the authors have not assessed expression of E2F2 protein or the DNA binding/transcriptional activity of E2F2 in the cells under study. The authors claim that their data "demonstrates that IFI6 is a new regulator of DNA replication". Of note, IFI6 knockout mice are normal and cells engineered to over-express do not show any obvious alterations in DNA synthesis. It is not clear where IFI6 is located in the cell or whether it is a part of the DNA replication machinery. Hence, there are no hints from previous publications, nor any compelling evidence from this manuscript, that "IFI6 is a new regulator of DNA replication". In sum, additional experiments would be needed to clarify the relationship between deregulated IFI6 expression, E2F control, accelerated DNA replication and the establishment of senescence.

3) Correlation between sensitivity to DNA replication stress-inducing drugs and pERK levels. The relative sensitivity of NRAS mutated melanoma cells to the cytotoxic effects of DNA replication stress-inducing drugs should be assessed compared to the levels of pERK1/2 signaling and IFI6 expression in the control (NRAS wild-type) cells. Since A375 cells express BRAF(V600E), the differences in sensitivity may not be related to pERK1/2 levels. This should be further investigated and clarified.

4) Lack of correlation between NRAS mutation status and IFI6 expression in the TCGA dataset. The authors conclude that mutational activation of NRAS in melanocytes drives expression of the known interferon-regulated genes IFI6, IFITM1 and MX1. However, this conclusion is not supported by analysis of TCGA data for cutaneous melanoma that comprises ~478 well curated specimens of which 278 specimens are informative for NRAS, BRAF or NF1 mutation and the expression of IFI6, IFITM1 and MX1 mRNAs. Using the MSKCC cBio Portal (http://www.cbioportal.org/public-portal/), one of the Reviewers interrogated this dataset and found no correlation between mutational activation of NRAS and the expression of the IFI6, IFITM1 and MX1 mRNAs. Indeed, mutational activation of NRAS and the expression of these mRNAs tends towards mutual exclusivity. The authors should address this concern, and the revised manuscript should discuss analysis of public datasets, and to what extent these analyses support (or do not) the overall molecular mechanism described here.

5) Missing experimental controls. The authors fail to provide an immunoblot that documents diminished IFI6 expression in the knockdown cells. Reviewers request a rescue experiment in which an RNAi resistant form of IFI6 rescues the knockdown phenotype with regard to colony formation in agarose, and if possible, tumorigenicity in mice. The authors fail to present an immunoblot that documents diminished E2F2 expression in the shE2F2 knockdown cells. Have the authors conducted specificity experiments to confirm that their siRNAs are working through E2F2? How was NF-κB knocked down in Figure 1? Please include an immunoblot that demonstrates knockdown of a component of NF-κB.

[Editors' note: further revisions were requested prior to acceptance, as described below.]

Thank you for resubmitting your work entitled "Interferon Α-Inducible Protein 6 regulates NRASQ61K-induced melanomagenesis and growth by regulating DNA replication" for further consideration at *eLife*. Your revised article has been favorably evaluated by Kevin Struhl as the Senior editor and a Reviewing editor.

The manuscript has been improved but there are some remaining issues that need to be addressed before acceptance, as outlined below:

Given the concerns raised by the reviewers about the indirect role of IFI6 in control of DNA replication, it is proposed that the title of the manuscript be changed from:

'Interferon Αlpha-Inducible Protein 6 regulates NRASQ61K-induced melanomagenesis and growth by regulating DNA replication'

to:

‘Interferon Αlpha-Inducible Protein 6 regulates NRASQ61K-induced melanomagenesis and growth’

Similarly, in the Abstract, the sentence:

'Collectively, we demonstrate that IFI6 via transcription factor E2F2 regulates DNA replication and melanoma development and growth and this pathway can be pharmacologically targeted to inhibit NRAS mutant melanoma.'

could be changed to:

'Collectively, we demonstrate that IFI6, via E2F2, regulates DNA replication and melanoma development and growth, and this pathway can be pharmacologically targeted to inhibit NRAS mutant melanoma.'

In the Introduction, the sentence:

'In addition, pharmacological inhibition of IFI6 with the MEK inhibitor trametinib, when combined with drugs that induce DNA replication stress'

should be changed to:

'In addition, treatment with the MEK inhibitor trametinib, which reduces IFI6 expression, when combined with drugs that induce DNA replication stress'

Results section, paragraph four, it should refer to Figure 2—figure supplement 2, not Figure 2—figure supplement 1. Authors make consider switching the order of these two figure supplements, so that they flow better with the text.

---

## [Author Response]

*The major concerns are:*

*1) Specificity for NRAS-driven tumors. The manuscript describes a mechanism that is supposedly specific for NRAS-driven melanomas, but Reviewers pointed out that the involvement of the RAF->MEK->ERK pathway indicates that melanomas driven by mutations in BRAF or silencing of NF1 should also display elevated IFI6 mRNA expression. This point should be carefully investigated and clarified. Is IFI6 upregulation unique to NRAS-driven tumors or not? Furthermore, by what mechanism(s) does NRAS-activated RAF->MEK->ERK activate NF-κB? Is it a direct mechanism of activation or does this occur through the secretion of inflammatory cytokines leading to NF-κB activation? If NRAS activates NF-κB signaling through the RAF->MEK->ERK pathway as described, then why wouldn't BRAF(V600E) or NF1(Null) melanomas, which have readily detectable pERK1/2 signaling, not also activate IFI6 mRNA expression? Indeed, it was noted that in their "NRAS wild-type" melanoma cell lines the authors employed at least one cell line (A375), which expresses BRAF(V600E) and has readily detectable pERK1/2 signaling. In sum, additional experiments are required to provide mechanistic insight into the mechanism of NRAS-mediated IFI6 expression through NF-κB activation and thus clarify the source of specificity (or lack thereof) for NRAS-driven melanoma.*

We thank the reviewers for asking these important questions. To address these concerns thoroughly, we performed several new experiments.

First, we showed that ectopic expression of BRAFV600E in MEL-ST cells, similar to NRASQ61K, results in the upregulation of IFI6 expression, which is consistent with the role of the MAPK pathway in the upregulation of IFI6. These new results are presented in Figure 1—figure supplement 3 of this revised manuscript.

Second, we knocked down the expression of NF1 in MEL-ST cells but did not observe an upregulation of IFI6. This indicates that there are still unknown effects of NF1 loss beyond MAPK pathway activation involved in the regulation of IFI6 expression. These results also show that NF1-deficient and NRAS-mutant melanoma are not equivalent, because NRASQ61K was able to activate IFI6, but loss of NF1 was not (also see our response to point 4 in this regard). These new results are presented in Figure 1—figure supplement 3.

The regulation of NF-κB by RAS proteins is a well-studied phenomenon. Therefore, to determine the mechanism of NF-κB upregulation, we first surveyed the literature and found that IKKβ is necessary for NF-κB activation in a mouse model of HRASv12-driven melanoma and for HRASv12-driven melanoma growth (see Yang et al., 2010, JCI PMID: 20530876). This study also showed that HRASv12 activates IKKβ kinase activity to activate NF-κB. IKKβ activates NF-κB via inflammatory cytokines, and in absence of IKKβ, inflammatory cytokines cannot activate NF-κB expression (see Li et al., 1999, J Exp. Med. PMID: 10359587 and Li et al., 1999, Science PMID: 10195897).

Therefore, we knocked down IKKβ expression in MEL-ST cells expressing NRASQ61K and asked if this inhibits NF-κB activity and the ability of NRASQ61K to activate IFI6. Similar to the results reported by Yang et al., we found that knockdown of IKKβ prevented NF-κB activation, as assessed by decreased IκBα phosphorylation, decreased NF-κB-responsive luciferase reporter activity, and decreased binding of NF-κB to the IFI6 promoter and the inability of NRASQ61K to activate IFI6 expression. These new results are presented in Figure 1—figure supplement 6 of this revised manuscript.

*2) Deregulated DNA replication versus senescence. The authors conclude that RNAi-mediated silencing of IFI6 leads to senescence, but this is not well documented and somewhat contradictory. The cells display elevated levels of SA-β-gal, a widely used but unreliable marker of senescence, and a decrease in H3K9 acetylation. However, the cells also display increased numbers of cells in S phase, which is contrary to the idea of senescence, an irreversible form of cell cycle arrest generally associated with arrest in G1. How can these observations be reconciled? Could the authors detect evidence of DNA damage such as collapsed DNA replication forks, gH2AX foci and/or the recruitment of RAD51 or 53BP1 into damage sites? Furthermore, in their initial gene induction studies and throughout the manuscript that authors employed Mel-ST cells, which are melanocytes immortalized by expression of hTERT and rendered susceptible to oncogenic transformation by expression of the SV40 Large and small T/t antigens. Since Large T binds to and inhibits the activity of the pRB, p107 and p130 "pocket proteins" that are negative regulators of E2F activity, and also to TP53, it is hard to understand mechanistically how senescence could be elicited in cells that are lacking these key tumor suppressors that are known to enforce the senescence program. Moreover, the authors claim that shIFI6 leads to elevated E2F2 mRNA expression (Figure 3) but since, to all intents and purposes, Mel-ST cells are TP53, pRB, p107 and p130 deficient, the significance of this observation is questionable since these cells should already have abundant E2F activity. It is also notable that the authors have not assessed expression of E2F2 protein or the DNA binding/transcriptional activity of E2F2 in the cells under study. The authors claim that their data "demonstrates that IFI6 is a new regulator of DNA replication". Of note, IFI6 knockout mice are normal and cells engineered to over-express do not show any obvious alterations in DNA synthesis. It is not clear where IFI6 is located in the cell or whether it is a part of the DNA replication machinery. Hence, there are no hints from previous publications, nor any compelling evidence from this manuscript, that "IFI6 is a new regulator of DNA replication". In sum, additional experiments would be needed to clarify the relationship between deregulated IFI6 expression, E2F control, accelerated DNA replication and the establishment of senescence.*

The second set of concerns also included several important questions. First, the reviewers mentioned that we did not document the senescence phenotypes very well. As the reviewers noticed, we used two established criteria (SA-β-gal assay and H3K9 acetylation) used by others to identify senescent cells. The use of H3K9Ac measurement was based on previous studies demonstrating increased heterochromatinization in senescent cells (for an example, see Narita et al., 2006, Cell PMID: 16901784).

Therefore, together, these markers are predictive of cellular senescence. As an additional DNA damage marker, we have now included p-H2A.X immunoblot. Regarding the collapsed DNA replication fork, in the previous version of our manuscript, we used the gold standard method for measuring DNA replication stress: the DNA combing assay. This assay clearly showed dysregulation of DNA replication, and, in conjunction with p- H2A.X, demonstrates DNA damage. These new results are presented in Figure 4, Figure 7, and Figure 8 of this revised manuscript.

The statement that senescence results in G1 phase cell cycle arrest is not entirely accurate, because the state of cell cycle arrest depends upon various factors including cell type and stimuli that induce senescence. For example, in cellular senescence, cells can be arrested in the G2/M phase (for a review on this topic, please refer to Gire and Dulic, 2015, Cell Cycle PMID: 25564883).

Second, regarding the statement that an increase of cells in S phase is counterintuitive for a senescence phenotype, we find that the loss of IFI6 results in E2F2-mediated unscheduled DNA replication (as demonstrated in the DNA combing assay, as well as the activation of several genes involved in DNA replication). As a result, cells incorporate more nucleotides, which causes a surge of cells in the S phase. Once this uncontrolled replication reaches a threshold level, the cell can no longer divide and therefore undergoes senescence. Please note that two previous studies have also documented this phenomenon in the context of cellular senescence. We cited these studies in our previous submission (Bartkova et al., 2006, Nature PMID: 17136093 and Di Micco et al., 2006 Nature PMID: 17136094) However, we understand the confusion this might have created for the reviewers; therefore, we have clearly articulated these results in the revised manuscript and added a paragraph in the Discussion section to clarify this point.

Third, we agree with the reviewers regarding the question of senescence induction in MEL-ST/NRASQ61K cells upon IFI6 knockdown. In the previous version of the manuscript, we did not test MEL-ST/NRASQ61K cells expressing IFI6 shRNA for cellular senescence induction, but tested only melanoma cell lines and primary human melanocytes. Therefore, we tested cellular senescence induction in MEL-ST cells transformed with NRASQ61K and expressing either IFI6 shRNA or nonspecific shRNA as a control. As suggested by the reviewers, we did not observe senescence in MEL-ST/NRASQ61K cells expressing IFI6 shRNA (see Figure 8—figure supplement 2). This is consistent with published studies demonstrating that the p53 and RB pathways are required for senescence induction.

We, then, asked what could be a possible reason for the attenuation of NRASQ61K-induced transformation upon IFI6 knockdown in MEL-ST/NRASQ61K cells. Previous studies have shown that uncontrolled DNA replication and consequent DNA damage can result in apoptosis induction in a p53- or RB-independent manner (for examples, see Strasser et al., 1994, PMID: 7954799; Knudsen et al., 2000, PMID: 11003670; and Aladjem et al., 1998, PMID: 9443911). To determine whether increased apoptosis induction might explain the observed phenotype in MEL-ST cells, we measured cell death and apoptosis using various assays (poly-HEMA coated plates, growth in detached condition assay (GILA), annexin V-FITC staining, and cleaved caspase 3 by immunoblotting). These assays were performed under detached conditions that mimic soft agar and in vivo tumor formation. We found that loss of IFI6 increased apoptosis induction in MEL-ST/NRASQ61K cells. These new results are presented in the Figure 8 of this revised manuscript. Taken together, these results show that loss of IFI6 expression results in DNA replication stress-induced DNA damage, which can cause senescence or apoptosis depending upon the presence or absence of active p53/RB proteins.

Finally, in regard to E2F gene activity/function, it is important not to generalize the functions of all E2F proteins. Previous studies have shown significant differences in these proteins, and context-specific functions for several of them have been described (for examples, see Attwooll et al., 2004, PMID: 15538380; DeGregori et al., 1997, PMID: 9207076; and DeGregori, 2002, Biochim Biophys Acta PMID: 12020800). It is also important to note that oncogenes can enhance the expression and/or activity of E2F family proteins (for an example, see Berkovich and Ginsberg, 2001, JBC PMID: 11551910).

However, we agree with the reviewers that it is important to show that E2F2 expression and function are enhanced after IFI6 knockdown. Therefore, we measured both RNA and protein level of E2F2 and our result showed that loss of IFI6 does result in increased E2F2 mRNA and protein expression (Figure 2—figure supplement 1). To show increased enrichment of E2F2 on the promoters of DNA replication regulatory genes that are upregulated in IFI6 knockdown cells, we performed ChIP experiments. Our results show increased enrichment of E2F2 on the MCM10 promoter after IFI6 knockdown. These new results are presented in Figure 4—figure supplement 2 of this revised manuscript.

In regard to the IFI6 knockout mice, we are not sure which study the reviewers are referring to. We do not find any evidence of IFI6 knockout mice in the literature. Moreover, none of our studies were performed using mouse cells, so we cannot comment on the role of IFI6 in mouse cells.

Regarding overexpression studies, we are not aware of any study that contradicts our results. Moreover, the fact that there is no hint of alteration in DNA replication in previous studies does not mean much unless a DNA replication stress phenotype was actively investigated. Moreover, none of the previous studies were performed in melanocytes or melanoma cells; therefore, it is hard to draw conclusions or determine the relevance of those previous studies in relation to our study.

Regarding, IFI6 being a regulator of DNA replication, we did not state that IFI6 directly regulates DNA replication by associating with the DNA replication machinery. Instead we showed that IFI6 loss results in the upregulation of E2F2, which in turn activates the expression of genes that are known regulators of DNA replication. Therefore, IFI6 loss results in uncontrolled DNA replication via upregulation of E2F2 and its target genes that regulate DNA replication, which is a more accurate description of our results and was discussed in our previous manuscript. Dysregulated DNA replication is one of the outcomes of IFI6 loss. Based on the questions from the reviewers, we have modified the language of our manuscript to more clearly show that the effect of IFI6 on DNA replication is not direct but rather mediated through E2F2.

*3) Correlation between sensitivity to DNA replication stress-inducing drugs and pERK levels. The relative sensitivity of NRAS mutated melanoma cells to the cytotoxic effects of DNA replication stress-inducing drugs should be assessed compared to the levels of pERK1/2 signaling and IFI6 expression in the control (NRAS wild-type) cells. Since A375 cells express BRAF(V600E), the differences in sensitivity may not be related to pERK1/2 levels. This should be further investigated and clarified.*

The above-mentioned point concerns the specificity of resistance towards the DNA replication stress-inducing agents. To further expand our results and determine the reason for this drug resistance, we performed additional experiments using additional BRAF-mutant and NF1-deficient melanoma cell lines. Our results reconfirmed our findings that NRAS-mutant melanoma cell lines were far more resistant to DNA replication stress-inducing drugs than other melanoma genotypes. These new results are presented in Figure 9 of this revised manuscript.

To determine whether IFI6 is upregulated by both oncogenic NRAS and oncogenic BRAF, and why only NRAS-mutant cell lines are resistant to DNA replication stress-inducing agents, we tested if the same set of genes is upregulated upon IFI6 knockdown in BRAF-mutant, NF1-deficient, and triple wild-type melanoma cells. We found that IFI6 knockdown did not result in increased expression of E2F2 or target genes that regulate DNA replication in BRAF-mutant, NF1-deficient, or triple wild-type melanoma cells, as was observed in NRAS-mutant melanoma cells. Thus, the sensitivity of melanoma cells to DNA replication stress-inducing drugs is not due to the ability of a genetic alteration (such as BRAFV600E) to simply activate IFI6 alone. Rather, it depends upon the ability of IFI6 to regulate DNA replication stress through E2F2-mediated transcription of key DNA replication target genes, which is seen only in the context of NRAS-mutant melanoma. These new results are presented in the Figure 9—figure supplement 1 of this revised manuscript.

*4) Lack of correlation between NRAS mutation status and IFI6 expression in the TCGA dataset. The authors conclude that mutational activation of NRAS in melanocytes drives expression of the known interferon-regulated genes IFI6, IFITM1 and MX1. However, this conclusion is not supported by analysis of TCGA data for cutaneous melanoma that comprises ~478 well curated specimens of which 278 specimens are informative for NRAS, BRAF or NF1 mutation and the expression of IFI6, IFITM1 and MX1 mRNAs. Using the MSKCC cBio Portal (http://www.cbioportal.org/public-portal/), one of the Reviewers interrogated this dataset and found no correlation between mutational activation of NRAS and the expression of the IFI6, IFITM1 and MX1 mRNAs. Indeed, mutational activation of NRAS and the expression of these mRNAs tends towards mutual exclusivity. The authors should address this concern, and the revised manuscript should discuss analysis of public datasets, and to what extent these analyses support (or do not) the overall molecular mechanism described here.*

Before describing our findings using the TCGA dataset, we would like to point out that in our previous submission, we presented results from several publically available datasets, in which we found that IFI6 expression was upregulated in melanoma samples compared to normal skin samples (see Figure 1). We also showed an analysis from a study that confirmed that IFI6 expression correlates with NRAS mutation status in clinical melanoma samples (see Figure 1).

We would also like to point out that based on the analysis of TCGA dataset there is no tendency for NRAS-mutant melanoma and expression of IFI6, IFIMT1, and MX1 to be mutually exclusive, because the p-values for these comparisons do not reach significance (p-value for IFI6 comparison in NRAS-mutant melanoma=0.555, p-value for IFI27 comparison in NRAS-mutant melanoma = 0.391, and p-value for MX1 comparison in NRAS-mutant melanoma =0.226).

Finally, and most importantly, we believe that the statement above assumes that the TCGA melanoma dataset is powerful enough to identify all MAPK target genes or BRAF/NRAS/NF1 signature genes. Thus, we put the TCGA melanoma dataset to this test. To this end, we selected two previously published studies and analyzed 71 genes in total. The first study identified 43 BRAF-MAPK-regulated target genes using BRAFV600E and MEK inhibitors (Joseph et al., 2010, PNAS PMID: 20668238), and the second study presented a large-scale analysis of patient-derived cutaneous melanoma samples that identified 29 genes that predict a mutant BRAF signature in melanoma (Kannengiesser et al., 2008 Molecular Oncology PMID: 19383316). Of the 43 genes from the Joseph et al. study, only 7 showed a significant tendency towards co-occurrence with the BRAFV600E mutation, and 5 showed a significant tendency towards mutual exclusivity with BRAFV600E. Surprisingly 31 of the 43 genes showed no correlation with the BRAFV600E mutation in the TCGA dataset.

Because in principal, as mentioned by the reviewers, oncogenic NRASQ61K and inactivation of NF1 should activate MAPK targets and correlate with these genotypes, we analyzed all 43 genes in the context of NRAS mutation or NF1 mutation/deletions. Our results showed that only 3 of the 43 genes showed a significant tendency towards co- occurrence in NRAS-mutant melanoma sample and 3 of the 43 genes showed a significant tendency towards mutual exclusivity, whereas 37 showed no correlation. Notably, these genes expect one did not show the same tendencies between the BRAF- and NRAS-mutant melanoma samples. In NF1-mutant melanomas, 3 of these 43 genes showed a significant tendency toward co-occurrence, and none of these 3 genes were common with NRAS-mutant and BRAF-mutant melanoma samples. Overall, this analysis showed that the TCGA data analysis identified less than 30% of MAPK targets in melanoma and less than 15% MAPK targets in NRAS- and less than 10% in NF1-mutant samples. Thus lack of correlation in the TCGA data between IFI6 expression and NRAS mutation should not be considered surprising based on this analysis.

The study by Kannengiesser et al. identified 29 genes that are upregulated in BRAF-mutant melanoma patient samples and thus represent a BRAF-mutant melanoma signature. Similar to our analysis with the Joseph et al. study, only 6 of the 2930 genes showed a significant tendency towards co-occurrence with BRAF-mutant melanoma. Thus, it failed to identify 23 of the 30 29 (796%) BRAF signature genes reported by Kannengiesser et al. Furthermore, only 1 gene showed a significant tendency towards co- occurrence with NRAS-mutant melanoma, and 7 showed a significant tendency toward mutual exclusivity. Finally, only 2 gene showed a significant tendency towards co- occurrence with NF1-deficient melanoma, and only one of these genes were identified to be significant for NRAS-mutant melanoma but not for BRAF-mutant melanoma.

Based on these preliminary analyses of the TCGA dataset we identified several genes that correlated with NRAS or NF1 mutations, but not with both genotypes, indicating differences between NRAS-mutant and NF1-deficient melanomas. This also provides a partial explanation for the differential regulation of IFI6 under NRAS-mutant and NF1-deficient conditions.

Thus, we conclude that datasets such as TCGA, although useful tools for researchers, are not comprehensive enough to predict or identify all biological outcomes in any given tumor type. We believe that the lack of correlation between IFI6 expression and NRAS mutation represents a limitation of the TCGA dataset rather than the proof that it is not regulated by the MAPK pathway. Therefore, independent studies, such as ours, should be encouraged and are complementary to the existing literature and large- scale datasets, including TCGA. We have included this analysis for the reviewers for review purpose as [Supplementary-material SD2-data] and would be happy to include it as part of our manuscript if the reviewers recommend that we do so.

*5) Missing experimental controls. The authors fail to provide an immunoblot that documents diminished IFI6 expression in the knockdown cells. Reviewers request a rescue experiment in which an RNAi resistant form of IFI6 rescues the knockdown phenotype with regard to colony formation in agarose, and if possible, tumorigenicity in mice. The authors fail to present an immunoblot that documents diminished E2F2 expression in the shE2F2 knockdown cells. Have the authors conducted specificity experiments to confirm that their siRNAs are working through E2F2? How was NF-κB knocked down in Figure 1? Please include an immunoblot that demonstrates knockdown of a component of NF-κB.*

As per the reviewers’ request, we included immunoblots that are in agreement with loss of IFI6 mRNA, confirming that IFI6 protein was also efficiently reduced in cells expressing IFI6 shRNA. These results are presented in Figure 2—figure supplement 1.

To show the specificity of the phenotype, we performed rescue experiments using two representative cell lines from our studies (YUGASP and MEL-ST/NRASQ61K). Our results show that ectopic expression of the IFI6 open reading frame (ORF) rescues growth in the soft agar assay and tumor growth in mice caused by IFI6 shRNA targeting the 3'-UTR. These new results are presented in Figure 2—figure supplement 2 and Figure 3—figure supplement 2.

We included data showing that E2F2 shRNA knocks down both E2F2 mRNA and protein levels. These new results are presented in Figure 5.

We also show that IFI6 loss activates E2F2 but not other E2F family genes. These new results are presented in Figure 2—figure supplement 1 and Figure 4—figure supplement 3. Furthermore, we show that E2F2 knockdown does not affect the expression of other E2F family genes, thus E2F2 shRNA is highly specific to E2F2. These new results are presented in Figure 6—figure supplement 1.

Regarding the NF-κB blot, NF-κB was knocked down by shRNA that targets p65 (RelA). We included an immunoblot to show that we were also able to knock down p65 at both the mRNA and protein levels. These new results are included in Figure 1—figure supplement 5.

[Editors' note: further revisions were requested prior to acceptance, as described below.]

*Given the concerns raised by the reviewers about the indirect role of IFI6 in control of DNA replication, it is proposed that the title of the manuscript be changed from:*

'Interferon Α-Inducible Protein 6 regulates NRASQ61K-induced melanomagenesis and growth by regulating DNA replication'

*to:*

*Interferon Α-Inducible Protein 6 regulates NRASQ61K-induced melanomagenesis and growth*

As suggested, we have changed the title to the recommended title.

*Similarly, in the Abstract, the sentence:*

*'Collectively, we demonstrate that IFI6 via transcription factor E2F2 regulates DNA replication and melanoma development and growth and this pathway can be pharmacologically targeted to inhibit NRAS mutant melanoma.'*

*could be changed to:*

*'Collectively, we demonstrate that IFI6, via E2F2, regulates DNA replication and melanoma development and growth, and this pathway can be pharmacologically targeted to inhibit NRAS mutant melanoma.'*

As suggested, we have changed the above-mentioned sentence to as recommended by the reviewers.

*In the Introduction, the sentence:*

*'In addition, pharmacological inhibition of IFI6 with the MEK inhibitor trametinib,when combined with drugs that induce DNA replication stress'*

*should be changed to:*

*'In addition, treatment with the MEK inhibitor trametinib, which reduces IFI6 expression, when combined with drugs that induce DNA replication stress'*

We have now modified the recommended sentence in Introduction to the sentence recommended by the reviewers.

*Results section, paragraph four, it should refer to Figure 2—figure supplement 2, not Figure 2—figure supplement 1. Authors make consider switching the order of these two figure supplements, so that they flow better with the text.*

As the reviewers may have noticed that the Figure 2—figure supplement 1 shows immunoblotting results for IFI6 knockdown and E2F2 expression in multiple cell lines. Because we wanted to test if IFI6 is required for NRASQ61K-induced transformation, we generated IFI6 knockdown in NRASQ61K transformed MEL-ST. Figure 2—figure supplement 1 shows effective IFI6 knockdown at protein level in this cell line. Please note that results illustrating reduced colony formation in soft agar after IFI6 knockdown is shown in Figure 2. Additionally, Figure 2—figure supplement 2 shows our results with the ectopic expression of shRNA-resistant IFI6 open reading frame (ORF) in NRASQ61K transformed MEL-ST expressing IFI6 shRNA. Therefore, there is no change required for the figure citations.